# Score-based Generative Modeling Secretly Minimizes the Wasserstein Distance

**Dohyun Kwon,   Ying Fan,   Kangwook Lee**
University of Wisconsin-Madison

## Abstract

Score-based generative models are shown to achieve remarkable empirical performances in various applications such as image generation and audio synthesis. However, a theoretical understanding of score-based diffusion models is still incomplete. Recently, Song et al. showed that the training objective of score-based generative models is equivalent to minimizing the Kullback–Leibler divergence of the generated distribution from the data distribution. In this work, we show that score-based models also minimize the Wasserstein distance between them under suitable assumptions on the model. Specifically, we prove that the Wasserstein distance is upper bounded by the square root of the objective function up to multiplicative constants and a fixed constant offset. Our proof is based on a novel application of the theory of optimal transport, which can be of independent interest to the society. Our numerical experiments support our findings. By analyzing our upper bounds, we provide a few techniques to obtain tighter upper bounds.

## 1   Introduction

The idea of gradually perturbing the data distribution to define the forward process and learning the reverse process has recently demonstrated remarkable performances in generative modeling. Score-based generative models [33, 34, 35] and diffusion probabilistic models [30, 17] have been used in numerous applications, including image generation [26, 12], and audio synthesis [9, 21].

Both model families have the following two processes. The given data distribution is incrementally perturbed by adding small noise in the forward process. This forward process can be understood as the standard diffusion process [35]. The reverse process requires the so-called *score function* [2], the gradient of the log probability density function. We minimize a weighted combination of score matching losses to approximate the score function. Once the score function is approximated, one can generate samples from randomly generated noise by applying the learned reverse process.

Note that score-based models do not explicitly minimize any sort of similarity between *the model distribution* (the distribution of the generated samples) and the data distribution. Therefore, it is unclear whether or not scored-based generative models can minimize divergence (or distance) between the model distribution and the data distribution. In very recent work [32], the authors show that optimizing the score matching loss is indeed equivalent to minimizing an upper bound on the Kullback–Leibler divergence (KL divergence). Therefore, it is expected that the model distribution is similar (in the sense of the KL divergence) to the data distribution if the score matching loss is small.

A natural question follows – *do score-based models minimize any other divergence or distance than the KL divergence?* At first glance, this seems unlikely. For instance, Generative Adversarial Networks (GANs) [16] were shown to minimize different divergences depending on the choice of the loss function. The original GAN loss function turns out to capture the Jensen–Shannon

---

Emails: dkwon7@wisc.edu, yfan87@wisc.edu, kangwook.lee@wisc.edu

36th Conference on Neural Information Processing Systems (NeurIPS 2022).

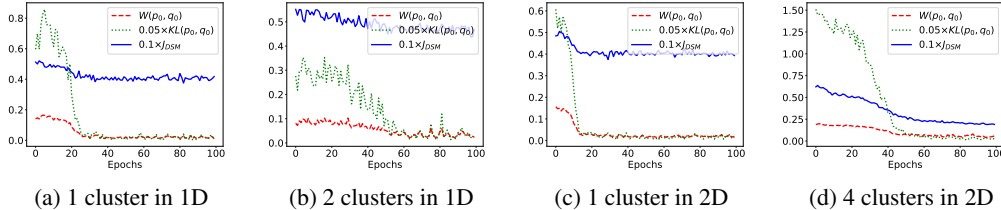

| (a) 1 cluster in 1D | (b) 2 clusters in 1D | (c) 1 cluster in 2D | (d) 4 clusters in 2D |

Figure 1: Training curves of loss function $J_{DSM}$ (eq. (20)), Wasserstein distance, and KL divergence during training. The training data is sampled from a mixture of 1D and 2D Gaussian clusters. (a): samples from one cluster $\mathcal{N}(0, 0.1)$. (b): samples from two clusters $\mathcal{N}(0, 0.1)$ and $\mathcal{N}(0.5, 0.1)$ with equal weights. (c): samples from one cluster $\mathcal{N}(\mathbf{0}, 0.01\mathbf{I})$. (d): samples from four clusters $\mathcal{N}((\pm 0.5, \pm 0.5)^\top, 0.01\mathbf{I})$ with equal weights.

divergence [16], while the W–GAN loss function captures the Wasserstein distance [3]. We refer the readers to [10] for more details.

On the other hand, a unique aspect of score-based models is that the score function is defined in the same space where data lies in. In stark contrast, all the other popular generative models, such as GANs [16], VAEs [19], and Normalizing Flows [20], use loss functions that have nothing to do with the underlying data space.

To gain more insights, we train score-based models on a few synthetic datasets. Shown in Fig. 1 are the training results. We plot the (scaled) training objectives in blue. Shown in green are the KL divergences, and one can note that the KL divergence decreases as the training objective decreases, as predicted in [32]. What we also observe is that as the training objective decreases, *the Wasserstein distance* also decreases. This is quite surprising as there is no direct relationship between the KL divergence and the Wasserstein distance, in general. Therefore, this phenomenon is not immediately implied by the fact that score-based models minimize the KL divergence.

Inspired by this observation, in this work, we prove the following property of score-based modeling:

> Score-based modeling minimizes the Wasserstein distance.

More specifically, using the theory of optimal transport, we investigate the evolution of data distribution based on partial differential equations (PDEs). This allows us to estimate the Wasserstein distance between the given data distribution and the distribution of the generated samples. Our work theoretically guarantees that the score-based generative models approximate data distributions in the space of probability measures equipped with the Wasserstein distance.

Our main contributions and findings are summarized as follows:

- (Theorem 1 & Corollary 1) Using the theory of optimal transport, we prove that the Wasserstein distance between the data distribution and the model distribution is bounded from above. The upper bound is given as the square root of the score-matching loss $J_{SM}$ up to multiplicative constants and a fixed constant offset. The main idea is to estimate the time-derivative of the Wasserstein metric by using the form of the continuity equations.

- (Theorem 2) We prove $J_{SM} \leq J_{DSM}$ (as defined in eq. (5) and eq. (20), respectively) under suitable assumptions, which gives us a more practical upper bound since $J_{DSM}$ is easier to estimate in practice and is widely used as a loss function to train score-based models.

- (Theorem 3) We analyze how the model distribution changes if the data distribution is perturbed initially. This guarantees that the score based-model is robust under noise.

- (Section 4) We corroborate our analysis by conducting numerical experiments. We also explore various techniques to tighten up the upper bound in practice.

**Notations:** $x = (x_1, x_2, \cdots, x_d)^\top \in \mathbb{R}^d$ and $t \in [0, T]$ denote the spatial variable and the time variable, respectively. Here, $\mathbb{R}^d$ is the $d$-dimensional Euclidean space and $T > 0$ is the time horizon. The gradient of a real-valued function $\rho$ with respect to the spatial variable and the time-derivative of $\rho$ are denoted by $\nabla \rho = \left( \frac{\partial \rho}{\partial x_1}, \frac{\partial \rho}{\partial x_2}, \cdots, \frac{\partial \rho}{\partial x_d} \right)^\top$, and $\partial_t \rho$, respectively. The Laplacian of $\rho$ is given as $\Delta \rho = \nabla \cdot (\nabla \rho)$. Here, $\nabla \cdot F = \frac{\partial F_1}{\partial x_1} + \frac{\partial F_2}{\partial x_2} + \cdots + \frac{\partial F_d}{\partial x_d}$ denotes the divergence of

$F = (F_1, F_2, \cdots, F_d)$ with respect to the spatial variable $x$. The Hessian of $\rho$ is a $d \times d$ matrix given by $(D^2 \rho)_{ij} = \frac{\partial^2 \rho}{\partial x_i \partial x_j}$ for $1 \leq i, j \leq d$. We indicate Gaussian distribution with mean $\mu$ and covariance $\Sigma$ by $\mathcal{N}(\mu, \Sigma)$, and $L_2$ norm by $\| \cdot \|$. We denote the data distribution and the distribution of the generated samples by $p_0$ and $q_0$.

## 2 Preliminaries and related works

In this section, we analyze the forward process and the reverse process in the score-based model through Partial differential equations (PDEs). In particular, we investigate the time evolution of marginal distributions through the two processes.

### 2.1 The forward process

As in [35], the forward process can be modeled by the standard diffusion process $\{x(t)\}_{t=0}^T$ for $t \in [0, T]$ such that $x(0) \sim p_0$ and

$$dx = f(x, t)dt + g(t)d\mathbf{w}. \tag{1}$$

Here, $\mathbf{w}$ is the standard Wiener process, $f : \mathbb{R}^d \times [0, T] \to \mathbb{R}^d$ is the drift coefficient of $x(t)$, and $g : [0, T] \to \mathbb{R}$ is the diffusion coefficient of $x(t)$.

Let $p_t = p(\cdot, t)$ be a probability density function of the random variable $x(t)$ on $\mathbb{R}^d$ at time $t \in [0, T]$. Then, the function $p : \mathbb{R}^d \times [0, T] \to [0, +\infty)$ satisfies the following partial differential equations, namely the Fokker–Planck equation or the forward Kolmogorov equation as in [5, Theorem 2.2]:

$$\partial_t p + \nabla \cdot (pf) - \frac{g^2}{2} \Delta p = 0, \ \ p(\cdot, 0) = p_0. \tag{2}$$

where $p_0$ represents the given data distribution. Under suitable assumptions on $f$ and $g$, a solution to the above initial value problem uniquely exists. We refer the readers to [6] for more details.

### 2.2 Understanding the reverse process through PDEs

To generate samples from $t = T$ to $0$, the score-based model uses the reverse-time SDE [2]:

$$dx = (f(x, t) - g(t)^2 \nabla \log p)dt + g(t)d\bar{\mathbf{w}} \tag{3}$$

Here, $\bar{\mathbf{w}}$ is a standard Wiener process when time flows backward from $t = T$ to $t = 0$.

As in the previous section, for given $x(T) \sim p_T$, a probability density function of the random variable $x(t)$ from the above process (3) satisfies the final value problem:

$$\partial_t p + \nabla \cdot \left( p \left( f - g^2 \nabla \log p \right) \right) + \frac{g^2}{2} \Delta p = 0, \ \ p(\cdot, T) = p_T. \tag{4}$$

Using this simple identity, $\Delta p = \nabla \cdot (\nabla p) = \nabla \cdot \left( p \frac{\nabla p}{p} \right) = \nabla \cdot (p \nabla (\log p))$, it can be seen that the differential equations in (2) and (4) are indeed identical.

### 2.3 The objective function

The score of the probability distribution, $\nabla \log p_t$, is approximated by a function $s_\theta : \mathbb{R}^d \times [0, T] \to \mathbb{R}^d$. For instance, in [32], the authors consider the weighted mean squared error (MSE):

$$J_{SM}(\theta; \lambda) := \frac{1}{2} \int_0^T \lambda(t) \mathbb{E}_{p_t} \left[ \| \nabla \log p_t(x) - s_\theta(x, t) \|^2 \right] dt \tag{5}$$

where $\lambda : [0, T] \to (0, \infty)$ is a positive weighting function.

Replacing $\nabla \log p$ in (4) with $s_\theta$, we deduce a new reverse process. For given noise distribution $q_T$, a probability density function $q_t = q(\cdot, t)$ at time $t \in [0, T]$ solves

$$\partial_t q + \nabla \cdot \left( q \left( f - g^2 s_\theta \right) \right) + \frac{g^2}{2} \Delta q = 0, \ \ q(T, \cdot) = q_T. \tag{6}$$

Here, we introduced a new function $q : \mathbb{R}^d \times [0, T] \to [0, +\infty)$ to illustrate the reverse process.

## 2.4 Related works

**Optimal transport**   The Wasserstein distance has a solid connection to the Fokker–Planck equation. In particular, the Fokker–Planck equation can be understood as the gradient flow in the space of probability measures equipped with the 2-Wasserstein distance [1, 18, 27]. For two probability measures $\mu$ and $\nu$ on $\mathbb{R}^d$, we define the 2-Wasserstein or Monge-Kantorovich distance as

$$W_2(\mu, \nu) := \inf \left\{ \int_{\mathbb{R}^d \times \mathbb{R}^d} \|x - y\|^2 d\gamma : \gamma \in \Pi(\mu, \nu) \right\}^{\frac{1}{2}}. \tag{7}$$

Here, $\Pi(\mu, \nu)$ is the set of all couplings $\pi$ of $\mu$ and $\nu$ where a Borel probability measure $\pi$ on $\mathbb{R}^d \times \mathbb{R}^d$ satisfies $\pi(A \times \mathbb{R}^d) = \mu(A)$ and $\pi(\mathbb{R}^d \times B) = \nu(B)$ for all measurable subsets $A, B \subset \mathbb{R}^d$.

Under suitable assumptions on the drift coefficient and the diffusion coefficient, a solution to the Fokker–Planck equation satisfies the so-called Wasserstein contraction property [7, 8, 29], which plays an important role in our analysis. More discussions can be found in Section 3.4 and Section 5.

**Score matching and diffusion probabilistic models**   Score matching models [33, 35] use the score function to reverse the noise corruption in a continuous time range by solving stochastic differential equations (SDEs). Diffusion probabilistic models [30, 17] adopt a similar idea of modeling the tractable denoising process to reverse each discrete step of noise corruption with probabilistic models. In fact, [35] show that diffusion models are score matching models with certain forms of SDE, which unifies the whole framework. Diffusion models are gaining increasing attention: recent advances [31, 26, 12] show that diffusion models can generate even better images than GANs while enjoying a more stable training process, which encourages it to be widely applied in various tasks, such as text-to-image and image editing [25, 24], audio generation [9, 21], etc.

# 3   Approximation in the Wasserstein space

In this section, we estimate the Wasserstein distance between a given data distribution $p_0$ and the marginal of the generated samples $q_0$ obtained by the score-based model. The proofs of the main results are deferred to Appendices A, B, and C. The key ideas are illustrated in Section 5.

## 3.1   Assumptions

We present our main results under the following assumptions.

**(A1)** The drift coefficient $f : \mathbb{R}^d \times [0, T] \to \mathbb{R}^d$ is Lipschitz continuous in the space variable $x$: there exists a positive constant $L_f(t) \in (0, \infty)$, depending on $t \in [0, T]$, such that for all $x, y \in \mathbb{R}^d$

$$\|f(x, t) - f(y, t)\| \leq L_f(t)\|x - y\|. \tag{8}$$

**(A2)** $s_\theta : \mathbb{R}^d \times [0, T] \to \mathbb{R}^d$ satisfies *the one-sided Lipschitz condition* [13, Definition 2.1]: there exists a constant $L_s(t) \in \mathbb{R}$, depending on $t \in [0, T]$, satisfying for all $x, y \in \mathbb{R}^d$

$$(s_\theta(x, t) - s_\theta(y, t)) \cdot (x - y) \leq L_s(t)\|x - y\|^2. \tag{9}$$

The usual Lipschitz constant of $s_\theta$ satisfies the above inequality (9). Unlike the usual Lipschitz constant, *the one-sided Lipschitz constant $L_s(t)$ is not necessarily nonnegative* (see Appendix D).

**(A3)** For the initial distribution $p_0$ of (2) and the final distribution $q_T$ of (6), the following quantities are finite:

$$\mathbb{E}_{p_0}[\|\log p_0\|], \quad \mathbb{E}_{p_0}[\Lambda(x)] \quad \mathbb{E}_{q_T}[\|\log q_T\|], \quad \text{and} \quad \mathbb{E}_{q_T}[\Lambda(x)] \tag{10}$$

where $\Lambda(x) := \log\max(\|x\|, 1)$.

**(A4)** There exist constants $C_1$ and $C_2$ such that

$$f(x, t) \cdot x \leq C_1\|x\|^2 + C_2 \text{ for all } x \in \mathbb{R}^d \text{ and } t \in [0, T]. \tag{11}$$

**(A5)** There exists a positive constant $C_3$ such that

$$\frac{1}{C_3} < g(t) < C_3 \text{ for all } t \in [0, T]. \tag{12}$$

**(A6)** $\int_0^T \mathbb{E}_{p_t}[f^2]dt$ and $\int_0^T \mathbb{E}_{q_t}[(f - g^2 s_\theta)^2]dt$ are finite.

The above assumptions guarantee the integrability of the score function $\nabla \log p$.

**Proposition 1.** *[6, Theorem 4.2.1] A solution $p_t$ to (2) satisfies $\nabla p_t \in L^1(\mathbb{R}^d)$ and*

$$\int_0^T \mathbb{E}_{p_t}\left[\|\nabla \log p_t(x)\|^2\right] dt < +\infty. \tag{13}$$

We give the proofs in smooth settings.

**(A7)** $p_0, q_T \in C^2(\mathbb{R}^d)$, and $g \in C^2([0, T])$. $f$ and $s_\theta$ are $C^2$ in both $x \in \mathbb{R}^d$ and $t \in [0, T]$.

**(A8)** A solution $p_t$ to (2) and a solution $q_t$ to (6) are positive $C^2$ probability densities decaying rapidly in $\mathbb{R}^d$: there exists $k > 0$ such that $p_t(x) = O(\exp(-\|x\|^k))$ and $q_t(x) = O(\exp(-\|x\|^k))$ for all $t \in [0, T]$. Furthermore, the vector fields $v[p_t] = f - \frac{1}{2}g^2 \nabla \log p_t$, and $v[q_t] = \left(f - g^2 s_\theta\right) + \frac{1}{2}g^2 \nabla \log q_t$ grow at most linearly as $\|x\| \to \infty$.

### 3.2 The main results

We now present a general result.

**Theorem 1.** *Let $p_0$ and $q_0$ be the data distribution and a pdf obtained by the score-based model as in Section 2. Then the Wasserstein distance between $p_0$ and $q_0$ is bounded as follows:*

$$W_2(p_0, q_0) \leq \int_0^T g(t)^2 I(t)\mathbb{E}_{p_t}\left[\|\nabla \log p_t(x) - s_\theta(x,t)\|^2\right]^{\frac{1}{2}} dt + I(T)W_2(p_T, q_T). \tag{14}$$

Here, a function $I : [0, T] \to (0, +\infty)$ defined by

$$I(t) := \exp\left(\int_0^t \left(L_f(r) + L_s(r)g(r)^2\right) dr\right) \tag{15}$$

is an integrating factor of the differential inequality (28) given later in Proposition 2. The main idea of the proofs is to estimate the derivatives of the Wasserstein distance along with the time evolution, which is illustrated in Section 5.

The first term on the right-hand side of (14) naturally arises from the estimation of $W_2(p_0, q_0)$. It is similar to a loss function $J_{SM}$ commonly used in the score-based models. Applying the Cauchy-Schwarz inequality, we obtain the upper bound in terms of the weighted MSE $J_{SM}$.

**Corollary 1.** *Let $p_0$ and $q_0$ be given in Theorem 1. If $\lambda(t) = g(t)^2$ for all $t \in [0, T]$, then the following inequality holds:*

$$W_2(p_0, q_0) \leq \sqrt{2\left(\int_0^T g(t)^2 I(t)^2 dt\right) J_{SM}} + I(T)W_2(p_T, q_T). \tag{16}$$

We defer the proofs of Theorem 1, and Corollary 1 to Appendix A.

**Remark 1.** *From the above result, the Wasserstein distance converges to zero as the loss function $J_{SM}$ goes to zero if $p_T = q_T$. However, due to the tension between $I(T)$ and $W_2(p_T, q_T)$, a general convergence result remains open (see Section 6 and Appendix B).*

**Remark 2.** *If $g^4 I^2 \lambda^{-1}$ is integrable in $[0, T]$, the similar inequality as in (34) holds for other choices of $\lambda$. See Corollary 4 in Appendix A.*

**Remark 3.** *Note that the above upper bound depends on the time horizon $T$. Numerical experiments for different choices of $T$ are provided in Section 4.3.*

We further investigate the offset $I(T)W_2(p_T, q_T)$ in Corollary 1. Consider the following case as in Denoising Diffusion Probabilistic Models (DDPM) [17]: for $\beta : [0, T] \to (0, +\infty)$, and $\sigma > 0$,

$$f(x, t) = -\frac{\beta(t)}{2} x \text{ and } g(t) = \sigma \sqrt{\beta(t)}. \tag{17}$$

In the above case, the stationary solution to (2) is given as the probability distribution of $\mathcal{N}(0, \sigma^2 I)$:

$$\phi(x) = \frac{1}{\sqrt{2\pi\sigma^2}} \exp\left(-\frac{\|x\|^2}{2\sigma^2}\right). \tag{18}$$

**Corollary 2.** *Under the same setting as in Corollary 1, let $q_T = \phi$. For $f, g$ given in (17), we have*

$$W_2(p_0, q_0) \leq \sqrt{2\left(\int_0^T \beta(t)\sigma^2 I(t)^2 dt\right) J_{SM}} + \exp\left(\int_0^T \beta(t)\sigma^2 L_s(t) dt\right) W_2(p_0, \phi). \tag{19}$$

The proof of Corollary 2 and the analysis of $L_s$ can be found in Appendix B.

**Remark 4.** *If $s_\theta$ is close enough to $\nabla \log p_t(x)$, then we have*

$$L_s(t) \sim (-\sigma^{-2} + \|D^2(\log h_t)\|_\infty).$$

*Here, $h_t := p_t/\phi$ exponentially converges to 1 as $t \to \infty$ (see Lemma 3). As a consequence, for sufficiently large $t > 0$, $\sigma^2 L_s(t)$ converges to $-1$, as observed in Fig. 4a. This yields that the second term in (19) decays as $T$ increases (see Fig. 4b).*

### 3.3 The relation between $J_{SM}$ and $J_{DSM}$

The above results give the guarantee that $J_{SM}$ can upper-bound Wasserstein distance. However, $J_{SM}$ is generally intractable due to the unknown quantity $\nabla_x \log p_t(x)$ [32], and is rarely used in practice. Alternatively, there is a conditional version of the score-matching loss $J_{DSM}$, which is equivalent to $J_{SM}$ up to a constant, as shown in [36]. $J_{DSM}$ is also widely used as a loss function to train score-based models:

$$J_{DSM}(\theta, \lambda) := \frac{1}{2} \int_0^T \lambda(t) \, \mathbb{E}_{p_0(x(0))p_{0t}(x|x(0))}[\|s_\theta(x, t) - \nabla_x \log p_{0t}(x|x(0))\|^2] dt, \tag{20}$$

where $p_{0t}$ indicates the conditional probability of $x$ at step $t$ given $x(0)$. Now we present our new result, which shows that $J_{DSM}$ is also an upper bound of $J_{SM}$ under a certain assumption.

**Theorem 2.** *If $p_{0t}$ satisfies*

$$\text{Var}[\mathbb{E}[(\nabla_x \log p_{0t}(x|x(0)))^\top | x(0)]] = 0, \tag{21}$$

*then we have*

$$J_{SM} \leq J_{DSM} \quad \text{and} \quad W_2(p_0, q_0) \leq \sqrt{2\left(\int_0^T g(t)^2 I(t)^2 dt\right) J_{DSM}} + I(T)W_2(p_T, q_T). \tag{22}$$

It is worth noting that Denoising Diffusion Probabilistic Models (DDPM) [17] satisfy (21).

**Corollary 3.** *Let $p(x(t)|x(t-1)) = \mathcal{N}(\sqrt{1-\beta_t}x(t-1), \beta_t I)$ as in Denoising Diffusion Probabilistic Models (DDPM) [17], which corresponds to $f(x, t) = -\frac{1}{2}\beta_t x$, and $g(t) = \sqrt{\beta_t}$. Here, $\beta_t = \beta(t) > 0$ is the noise schedule for all $t$, and let $\lambda(t) = g(t)^2$. Then, (22) holds.*

The proofs of Theorem 2 and Corollary 3 can be found in Appendix C. We also provide numerical results which show $J_{SM} \leq J_{DSM}$ for DDPM in Appendix F.

**Remark 5.** *Using Theorem 2 and Corollary 3, we can present an upper bound of the Wasserstein distance in terms of $J_{DSM}$, where both $W_2(p_0, q_0)$ and $J_{DSM}$ are tractable. This enables us to empirically verify this upper bound via DDPM in Section 4.*

## 3.4 Perturbation and sensitivity analysis

Let $\tilde{p}_0$ be a small perturbation of the original data distribution $p_0$. For a given corrupted distribution $\tilde{p}_0$, denote the probability density functions in the forward process and the reverse process by $\tilde{p}_t$ and $\tilde{q}_t$, respectively. We investigate how this perturbation changes the score-based model. In particular, we are interested in the upper bound on $W_2(q_0, \tilde{q}_0)$ with respect to $W_2(p_0, \tilde{p}_0)$.

It is well known from the Wasserstein contraction property [8] that the Wasserstein distance between two probability distributions in the forward process can be estimated as follows: for all $t \in [0, T]$

$$W_2(p_t, \tilde{p}_t) \leq \exp\left( \int_0^t L_f(r) dr \right) W_2(p_0, \tilde{p}_0). \tag{23}$$

In the reverse process, the score function in the drift coefficients may vary, and thus the above inequality does not directly give us the upper bound on $W_2(q_0, \tilde{q}_0)$. Instead, our analysis in the previous sections allows us to estimate $W_2(q_0, \tilde{q}_0)$.

It is worth noting that the Wasserstein distance satisfies the triangle inequality, unlike other statistical measures, including the Kullback–Leibler (KL) divergence. Combining our main results with

$$W_2(q_0, \tilde{q}_0) \leq W_2(p_0, \tilde{p}_0) + W_2(p_0, q_0) + W_2(\tilde{p}_0, \tilde{q}_0),$$

we conclude the following theorem.

**Theorem 3.** *Under the same setting as in Corollary 1, let $\tilde{p}_0$ be the corrupted data distribution and $\tilde{q}_0$ be a probability density function obtained by the score-based model starting from $\tilde{p}_0$. It holds that*

$$W_2(q_0, \tilde{q}_0) \leq W_2(p_0, \tilde{p}_0) + \sqrt{2\left( \int_0^T g(t)^2 I(t)^2 dt \right) J_{SM}} + \sqrt{2\left( \int_0^T g(t)^2 \tilde{I}(t)^2 dt \right) \tilde{J}_{SM}}$$
$$+ \tilde{I}(T) W_2(p_T, q_T) + \tilde{I}(T) W_2(\tilde{p}_T, \tilde{q}_T). \tag{24}$$

Here, $\tilde{J}_{SM}$ and $\tilde{I}$ denote the weighted MSE and the integrating factor from the perturbed model.

**Remark 6.** *The Wasserstein distance between the original data distribution $p_0$ and the model distribution $\tilde{q}_0$ starting from the corrupted data $\tilde{p}_0$ can also be estimated similarly.*

## 4 Experiments

Here we present experiments on simulation datasets to verify the upper bound in eq. (22). Code is available at `https://github.com/UW-Madison-Lee-Lab/score-wasserstein`.

### 4.1 Experiment settings

**Datasets** Here we adopt three 2D datasets for simulation: One cluster Gaussian $\mathcal{N}(\mathbf{0}, 0.1\mathbf{I})$, two moons in [28], and four clusters Gaussian mixture $\mathcal{N}((\pm 0.5, \pm 0.5)^\top, 0.01\mathbf{I})$ with equal weights for each cluster. Visualization for these datasets is in Fig 2.

For forward dynamics, we use discrete timesteps with $T = 10$. For $t \in [1, 10]$, $p(x(t)|x(t-1)) = \mathcal{N}(\sqrt{1 - \beta_t} x(t-1), \beta_t I)$ as in DDPM [17], i.e., $f(x, t) = -\frac{1}{2}\beta_t x$, and $g(t) = \sqrt{\beta_t}$. For $\beta_t$, we adopt a sigmoid schedule from $\beta_1 = 10^{-5}$ to $\beta_T = 10^{-2}$.

**Training and evaluation** We use a 4-layer neural network as the score matching model, with ReLU nonlinearity and skip-connection at the final output. Each layer is composed of a linear layer with 64 hidden neurons and an embedding layer for 10 timesteps. For optimizer, we use AdamW [22] with learning rate $= 0.001$ and weight decay coefficient $0.01$. To evaluate Wasserstein distance, we use POT [15] to compute $W_2(p_0, q_0)$ at the end of each training epoch. To avoid the term of $W_2(p_T, q_T)$ in eq. (22) for verification, we first perform forward pass on sampled $x(0)$ to get $x(T)$, and perform data generation on the same $x(T)$ to fix $q_T = p_T$ and $W_2(p_T, q_T) = 0$. For loss function, we use $J_{DSM}$ with $\lambda(t) = g(t)^2$ and batch size $= 128$. To get both lower and higher losses to verify our bound, we first maximize the loss for 10 epochs and then minimize the loss till convergence.

## 4.2 Results

### 4.2.1 The upper bound

To see the upper bound in eq. (22), we take the log on each side of the equation with $W_2(p_T, q_T) = 0$:

$$\log W_2(p_0, q_0) \le \underbrace{\frac{1}{2} \log J_{DSM}}_{\text{slope}} + \underbrace{\frac{1}{2} \log \left( \int_0^T g(t)^2 I(t)^2 dt \right)}_{\text{intercept}}. \tag{25}$$

We present a log-log plot to verify the slope and the intercept in the upper bound respectively while fixing $p_T = q_T$: see Fig 2. The log-log plots are obtained by evaluating Wasserstein distance and the loss function during training. The red dots are the sampled data from each dataset, the green dots are generated data after loss maximization, and the purple dots are the generated data sampled at convergence. From the log-log plots, we can observe that the theoretical line (the black line) is a valid upper bound of the empirical one (the blue dots).

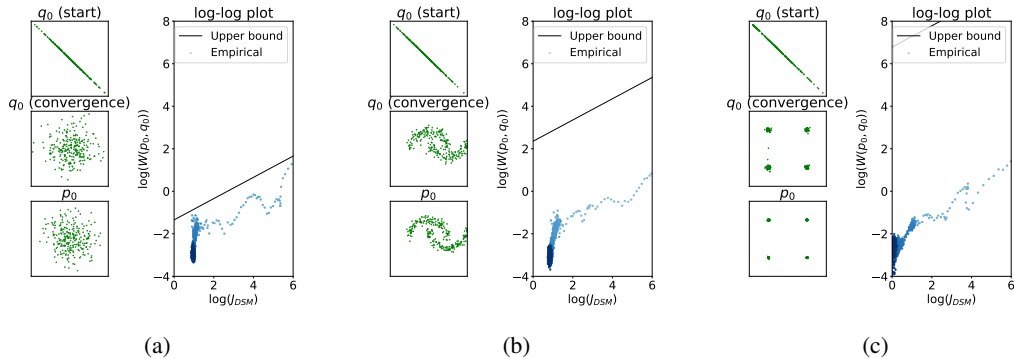

Figure 2: Log-log plots of the loss function and Wasserstein distance, with the corresponding generated data at the start of the loss minimization and at convergence for each dataset. The darkness of the data points corresponds to training epochs: the darker ones are closer to convergence.

### 4.2.2 Effect of weight regularization

Here we explore weight constraints that can affect the tightness of the theoretical upper bound while fixing $p_T = q_T$. As shown in Fig 2, the gap between the theoretical bound and the empirical one mainly results from the intercept in eq. (25). Recall our intercept is $\frac{1}{2} \log \left( \int_0^T g(t)^2 I(t)^2 dt \right)$ for $I(t)$ given in (15). Since $g(t)$ and $L_f$ are fixed, only the one-sided Lipchitz constant $L_s(r)$ of the model $s_\theta$ is subject to the training process, which is estimated via grid search in our experiments.

Notice that the loss function $J_{DSM}$ aims to minimize the distance between $s_\theta(x, t)$ and $\nabla_x \log p_{0t}(x|x(0))$. Thus, $L_s$ at convergence is determined by dynamics $\nabla_x \log p_{0t}(x|x(0))$. If the one-sided Lipschitz constant for $\nabla_x \log p_{0t}(x|x(0))$ is small, then the intercept would be expected to be small (see example in Fig 2a of Gaussian cluster with moderately high variance). If the one-sided Lipschitz constant for $\nabla_x \log p_{0t}(x|x(0))$ is large, to minimize the loss, the intercept would also be large (see example in Fig 2c of Gaussian clusters with lower variance). However, it can be possible to make the upper bound tighter while sacrificing the training loss by constraining $L_s$.

To constrain $L_s$, we adopt spectral normalization [4], weight clipping with threshold = 0.1[3]. As we can observe in Fig. 3, loss at convergence will increase when we impose very strong regularization on the weights while the intercept decreases. Importantly, the intercept approaches the empirical one without not exceeding it. However, if the regularization is too strong, the training loss at convergence would also become larger, resulting in worse quality of the generated data. See more examples of weight decay in Appendix G, where we also present the log-log training plots with different weight decay coefficients.

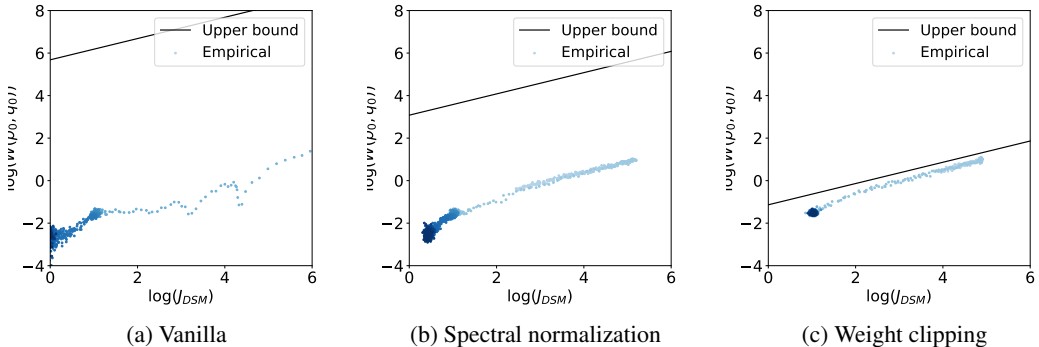

| (a) Vanilla | (b) Spectral normalization | (c) Weight clipping |

Figure 3: Effect of weight clipping and spectral normalization.

### 4.3 Effect of the number of total time steps: the decay of the offset

In our Remark 4, we argue that the offset in our upper bound would decay to $0$ when choosing the appropriate forward SDE and sufficiently large $T$. Here we provide empirical evidence for our claim *without* fixing $p_T = q_T$. We use the four clusters Gaussian dataset and fix $\beta$ to be a sigmoid schedule in the range of $[10^{-5}, 10^{-2}]$ for all choices of $T$. In Fig. 4, we plot $L_s(t)$ for $T = 100$ to show that $L_s(t)$ would converge to $-1$ when $t \to T$. Moreover, since $W_2(p_T, q_T)$ decays exponentially (see eq. 39), we also show that as $T$ increases, our offset $I(T)W_2(p_T, q_T)$ would also decay in a rate that is approximately exponential when $T$ is sufficiently large, as the model converges to the actual data distribution ($W_2(p_0, q_0) \approx 0$).

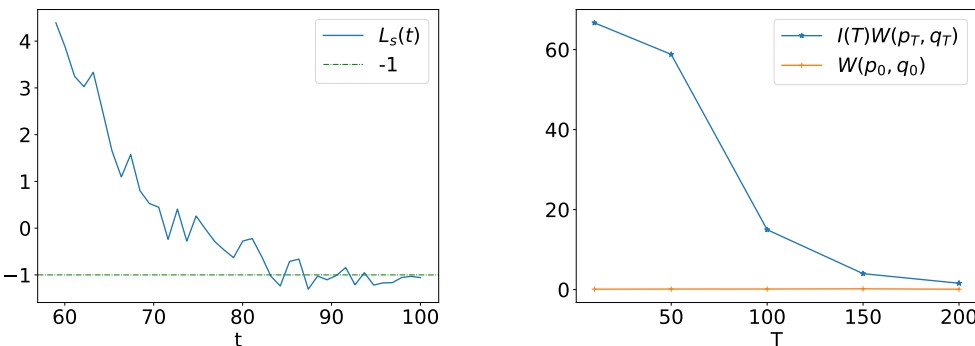

(a) $L_s(t)$ for $T = 100$. We can verify that when $T$ is sufficiently large, $L_s(T) \to -1$ as suggested in Remark 4. $L_s(t)$ when $t \le 50$ is cut due to visibility of the threshold 0 and -1. The full figure is in Appendix E.

(b) $I_T W_2(p_T, q_T)$ and $W_2(p_0, q_0)$ for $T = 10, 50, 100, 150, 200$ respectively at the convergence of training, while $\beta$ to be sigmoid schedule from $\beta_1 = 10^{-5}$ to $\beta_T = 10^{-2}$ for all choices of $T$.

Figure 4: Effect of $T$. We can observe that $L_s(T)$ would converge to $-1$ and $I(T)W_2(p_T, q_T)$ would decay almost exponentially when $T$ is sufficiently large ($\ge 100$).

## 5 The main idea of the proofs

We investigate the time evolution of probability measures arising in the score-based models. Using the optimal transport theory, we differentiate the Wasserstein distances, $W_2(p_t, q_t)$ with respect to time. The main idea is to express these equations (2) and (6) in the form of the continuity equation:

$$\partial_t \rho + \nabla \cdot (\rho v) = 0 \qquad (26)$$

where $v$ is a so-called velocity field. Let us denote the velocity fields of $p$, and $q$ by

$$v[p] = f - \frac{1}{2}g^2 \nabla \log p, \text{ and } v[q] = \left(f - g^2 s_\theta\right) + \frac{1}{2}g^2 \nabla \log q.$$

Then, the equations (2), and (6) can be written in the form of (26) with $v[p]$ and $v[q]$, respectively.

Using these vector fields, we compute the time derivative of the Wasserstein distance between two evolving probability measures. For velocity fields $v[\rho^{(i)}]$ depending on $\rho^{(i)}$, let $\rho^{(i)}$ for $i = 1, 2$ be two solutions of the continuity equations

$$\partial_t \rho^{(i)} + \nabla \cdot (\rho^{(i)} v[\rho^{(i)}]) = 0.$$

Then, the relation between the Wasserstein distance and velocity fields arises from the theory of optimal transport:

$$\frac{1}{2}\frac{d}{dt}W_2^2(\rho_t^{(1)}, \rho_t^{(2)}) = \mathbb{E}_{\pi_t}\left[(x - y) \cdot (v[\rho_t^{(1)}](x) - v[\rho_t^{(2)}](y))\right] \tag{27}$$

where $\pi_t$ is an optimal transport plan between $\rho_t^{(1)}$ and $\rho_t^{(2)}$. See [1, Theorem 8.4.7] and [29, Corollary 5.25]. This equality plays an important role when we estimate the Wasserstein distance. The following proposition is a consequence of (27) and Lemma 1 in Appendix A.2.

**Proposition 2.** *Let $p_t$ and $q_t$ be solutions to* (2) *and* (6), *respectively, in* $[0, T]$. *Then,*

$$-\frac{d}{dt}W_2(p_t, q_t) \le (L_f + L_s g^2)W_2(p_t, q_t) + g^2 b^{\frac{1}{2}} \tag{28}$$

*where* $b(t) := \mathbb{E}_{p_t}\left[\|\nabla \log p_t(x) - s_\theta(x, t)\|^2\right]$.

## 6 Discussions

**Convergence of the Wasserstein distance**    If $p_T = q_T$, then our result yields that the Wasserstein distance $W_2(p_0, q_0)$ converges to zero as $J_{SM}$ goes to zero. However, if $p_T$ is not equal to $q_T$ and $f$ and $g$ are not explicitly given, then we find that it is not easy to obtain such a convergence result. This is due to the nature of the Langevin dynamics in the reverse process. Typically, additional assumptions such as the log-concavity are required to guarantee the Langevin dynamics convergence, for instance, [11]. In addition, the dependence of the neural network $s_\theta$ makes it hard to evaluate the precise upper bound. We expect that appropriate assumptions on $s_\theta$ and the initial distribution $p_0$ may resolve this issue, but we do not pursue them in this paper.

**Estimating $L_s$**    Estimating our theoretical bound requires estimating the one-sided Lipschitz constant $L_s$, which is NP-hard [37], making it difficult to estimate in high dimensional data. Also, estimating the one-sided Lipschitz constant does not enjoy the semi-definiteness of estimating the two-sided one in [14]. Note that the two-sided Lipschitz constant is an upper bound of the one-sided one, which can be more efficiently estimated as shown in [14]. However, it would result in a looser theoretical upper bound if we adopt the two-sided Lipschitz constant.

## 7 Conclusion

In this paper, we mainly discuss the relationship between the loss function of score-based models and the Wasserstein distance. With the help of the theory in optimal transport, we present a novel upper bound on Wasserstein distance in terms of the loss function used in practice. We also provide numerical experiments to verify our theoretical bound with various simulation datasets. Moreover, we explore factors that affect the estimation of the upper bound and show the corresponding influence on the training process.

## Acknowledgments and Disclosure of Funding

Support for this research was provided by the University of Wisconsin-Madison Office of the Vice Chancellor for Research and Graduate Education with funding from the Wisconsin Alumni Research Foundation, and NSF Award DMS-2023239.

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
