# A  Proofs of Theorem 1 and Corollary 1

## A.1  Proofs

*Proof of Theorem 1.*  Recall the differential inequality (28) below from Proposition 2:

$$-\frac{d}{dt}W_2(p_t, q_t) \le (L_f + L_s g^2)W_2(p_t, q_t) + g^2 b^{\frac{1}{2}}.$$

It can be solved by introducing the integrating factor,

$$I(t) := \exp\left(\int_0^t L_f(r) + L_s(r)g(r)^2 dr\right) \text{ where } b(t) := \mathbb{E}_{p_t}\left[\|\nabla \log p_t(x) - s_\theta(x,t)\|^2\right]. \quad (29)$$

As $\frac{d}{dt}I(t) = (L_f(t) + L_s(t)g(t)^2)I(t)$, the above inequality (28) can be written as

$$-\frac{d}{dt}\left\{I(t)W_2(p_t, q_t)\right\} \le g(t)^2 I(t)b(t)^{\frac{1}{2}}.$$

Integrating both sides from $0$ to $T$, we obtain that

$$I(0)W_2(p_0, q_0) - I(T)W_2(p_T, q_T) \le \int_0^T g(t)^2 I(t)b(t)^{\frac{1}{2}} dt.$$

As $I(0) = 1$, we conclude that

$$W_2(p_0, q_0) \le \int_0^T g(t)^2 I(t)b(t)^{\frac{1}{2}} dt + I(T)W_2(p_T, q_T). \quad (30)$$

$\square$

*Proof of Corollary 1.*  Let

$$J_I(\theta) = \int_0^T g(t)^2 I(t)b(t)^{\frac{1}{2}} dt. \quad (31)$$

Here, $I(t)$ and $b(t) = \mathbb{E}_{p_t}\left[\|\nabla \log p_t(x) - s_\theta(x,t)\|^2\right]$ are given in (29).  The Cauchy-Schwarz inequality yields

$$J_I(\theta) \le \left(\int_0^T 2g(t)^4 I(t)^2 \lambda(t)^{-1} dt\right)^{\frac{1}{2}} \left(\frac{1}{2}\int_0^T \lambda(t)b(t)dt\right)^{\frac{1}{2}}. \quad (32)$$

Since $\lambda = g^2$ and $J_{SM}$ given in (5) satisfies

$$J_{SM}(\theta; \lambda) = \frac{1}{2}\int_0^T \lambda(t)\mathbb{E}_{p_t}\left[\|\nabla \log p_t(x) - s_\theta(x,t)\|_2^2\right] dt = \frac{1}{2}\int_0^T \lambda(t)b(t)dt, \quad (33)$$

we conclude (34):

$$W_2(p_0, q_0) \le \sqrt{2\left(\int_0^T g(t)^2 I(t)^2 dt\right) J_{SM}} + I(T)W_2(p_T, q_T).$$

$\square$

**Remark 7.** *It is expected that the parallel results hold for weak solutions to (2) and (4) by using suitable approximations. This approach has been studied for the Wasserstein contraction property in [8, Section 6.2].*

From (32) and (33) in the proof of Corollary 1, we obtain the following result for other choices of $\lambda$.

**Corollary 4.** *Let $p_0$ and $q_0$ be given in Theorem 1. Suppose that $g^4 I^2 \lambda^{-1}$ is integrable in $[0, T]$. Then the following inequality holds:*

$$W_2(p_0, q_0) \le \sqrt{2\left(\int_0^T g(t)^4 I(t)^2 \lambda(t)^{-1} dt\right) J_{SM}} + I(T)W_2(p_T, q_T). \quad (34)$$

## A.2 Technical lemmas

**Lemma 1.** *Let $\pi_t$ be an optimal transport plan between $p_t$ and $q_t$. Then, we have*

$$\mathbb{E}_{\pi_t}\left[(x-y)\cdot(v[q_t](y)-v[p_t](x))\right] \le W_2(p_t,q_t)\left\{(L_f+L_s g^2)W_2(p_t,q_t)+g^2 b^{\frac{1}{2}}\right\} \quad (35)$$

*where $b(t) := \mathbb{E}_{p_t}\left[\|\nabla\log p_t(x) - s_\theta(x,t)\|^2\right].$*

*Proof.* The left-hand side of (35) is given by

$$\mathbb{E}_{\pi_t}\left[(x-y)\cdot(v[q_t](y)-v[p_t](x))\right] = \mathbb{E}_{\pi_t}\left[(x-y)\cdot(f(y,t)-f(x,t))\right]$$
$$+ g^2\mathbb{E}_{\pi_t}\left[(x-y)\cdot(\nabla\log p_t(x)-s_\theta(y,t))\right]$$
$$+ \frac{g^2}{2}\mathbb{E}_{\pi_t}\left[(x-y)\cdot(\nabla\log q_t(y)-\nabla\log p_t(x))\right].$$

In Lemma 2 below, we prove that the last term is less than or equal to zero.

In what follows, we estimate the first two terms. First, using the Lipschitzness of $f$ in space, we get

$$\mathbb{E}_{\pi_t}\left[(x-y)\cdot(f(y,t)-f(x,t))\right] \le L_f\mathbb{E}_{\pi_t}\left[\|x-y\|^2\right] = L_f W_2^2(p_t,q_t).$$

The last equality follows from the fact that $\pi_t$ is an optimal plan between $p_t$ and $q_t$.

Next, the second term $g^2\mathbb{E}_{\pi_t}\left[(x-y)\cdot(\nabla\log p_t(x)-s_\theta(y,t))\right]$ is the sum of the following two terms:

$$I_1 := g^2\mathbb{E}_{\pi_t}\left[(x-y)\cdot(s_\theta(x,t)-s_\theta(y,t))\right]$$

and

$$I_2 := g^2\mathbb{E}_{\pi_t}\left[(x-y)\cdot(\nabla\log p_t(x)-s_\theta(x,t))\right].$$

As shown above, the former one $I_1$ is bounded from above by $g^2 L_s W_2^2(p_t,q_t)$.

It suffices to find an upper bound on the latter one $I_2$. By the Cauchy-Schwarz inequality, we have

$$I_2 \le g^2\mathbb{E}_{\pi_t}\left[\|x-y\|^2\right]^{\frac{1}{2}}\mathbb{E}_{\pi_t}\left[|\nabla\log p_t(x)-s_\theta(x,t)|^2\right]^{\frac{1}{2}}.$$

As the marginals of $\pi_t$ are $p_t$ and $q_t$, it holds that

$$\mathbb{E}_{\pi_t}\left[\|\nabla\log p_t(x)-s_\theta(x,t)\|^2\right] = \mathbb{E}_{p_t}\left[\|\nabla\log p_t(x)-s_\theta(x,t)\|^2\right].$$

As a consequence, we conclude that

$$I_1 + I_2 \le g(t)^2 W_2(p_t,q_t)\left\{L_s W_2(p_t,q_t)+b(t)^{\frac{1}{2}}\right\}$$

where $b(t) = \mathbb{E}_{p_t}\left[\|\nabla\log p_t(x)-s_\theta(x,t)\|^2\right].$ $\qquad\square$

Before proving the lemma below, let us recall some basic definitions from the theory of optimal transport. The Wasserstein distance defined in (7) has an equivalent formulation:

$$W_2(\mu,\nu) = \inf\left\{\int_{\mathbb{R}^d}\|x-T(x)\|^2 d\mu : T_\#\mu = \nu\right\}^{\frac{1}{2}}. \quad (36)$$

The optimizer of the above problem is called the optimal map from $\mu$ to $\nu$. It is well known that there exists a convex function $\phi$ such that $T = \nabla\phi$.

**Lemma 2.** $\mathbb{E}_{\pi_t}\left[(x-y)\cdot(\nabla\log q_t(y)-\nabla\log p_t(x))\right]$ *is nonpositive.*

*Proof.* Let $T_t$ be an optimal transport map from $p_t$ to $q_t$ and a convex function $\phi_t$ satisfy $\nabla\phi_t = T_t$ for all $t \in [0,T]$. As in the proof of [7, Theorem 1], we have

$$\mathbb{E}_{\pi_t}\left[(x-y)\cdot(\nabla\log q_t(y)-\nabla\log p_t(x))\right] = -\mathbb{E}_{p_t}[\Delta\phi_t+\Delta\phi_t^*(\nabla\phi_t)-2d] \quad (37)$$

where $\phi_t^*$ is a convex conjugate of $\phi_t$. The convexity of $\phi_t$ yields that $\Delta\phi_t+\Delta\phi_t^*(\nabla\phi_t)-2d$ and we conclude. $\qquad\square$

# B  Further analysis of the upper bound

*Proof of Corollary 2.* Recall from Corollary 1 that

$$W_2(p_0, q_0) \leq \sqrt{2 \left( \int_0^T g(t)^2 I(t)^2 dt \right) J_{SM} + I(T) W_2(p_T, q_T)}. \tag{38}$$

Based on the contraction property [8], we quantify the Wasserstein distance between $p_T$ and $\phi$.

$$W_2(p_T, \phi) \leq \exp\left( -\int_0^T \frac{\beta(t)}{2} dt \right) W_2(p_0, \phi). \tag{39}$$

Using the above, the definition of $I(t)$, and $q_T = \phi$, we conclude (19). □

It worth noting that as a consequence of (39), $W_2(p_T, \phi)$ is small for an appropriate choice of $T$ and $\beta(t)$.

## B.1  Exponential convergence of $h_t$

For simplicity of notations, we define the norm in $L^2(\phi)$ as follows,

$$\|f\|_{L^2(\phi)} := \left( \int_{\mathbb{R}^d} f^2 d\phi \right)^{\frac{1}{2}}, \tag{40}$$

where $\phi$ is given in (18). In addition, assume that

$$\beta(t) > c > 0 \text{ for all } t \geq 0. \tag{41}$$

**Lemma 3.** *Under the same setting as in Corollary 2, we have*

$$\|h_t - 1\|_{L^2(\phi)} \leq \exp\left( -\frac{\sigma^2 \lambda}{2} \int_0^T \beta(t) dt \right) \|h_0 - 1\|_{L^2(\phi)}. \tag{42}$$

*where $h_t = p_t/\phi$ for some constant $\lambda > 0$. In particular, $h_t$ exponentially converges to 1 in $L^2(\phi)$ for $\beta$ satisfying (41).*

For a constant function $\beta$, Lemma 3 is proven in [23]. For the sake of completeness, we provide the proof, which is a small modification of [23, Section 2].

*Proof of Lemma 3.* In our case, the equation (2) of $p_t$ is given by

$$\partial_t p - \frac{\beta}{2} \left( \nabla \cdot (px) + \sigma^2 \Delta p \right) = 0, \quad p(\cdot, 0) = p_0. \tag{43}$$

By direct computations, we obtain the equation of $h(x, t) = h_t(x)$ as follows:

$$\partial_t h - \frac{\beta}{2} \left( -\nabla h \cdot x + \sigma^2 \Delta h \right) = 0, \quad h(\cdot, 0) = h_0. \tag{44}$$

We estimate $\|h - 1\|_{L^2(\phi)}^2$ by differentiating it with respect to time. Using the integration by parts and Poincaré inequality, we obtain that

$$\frac{d}{dt} \|h - 1\|_{L^2(\phi)}^2 = -\sigma^2 \beta(t) \|\nabla h\|_{L^2(\phi)}^2 \leq -\sigma^2 \lambda \beta(t) \|h - 1\|_{L^2(\phi)}^2 \tag{45}$$

This yields (42). Lastly, for $\beta$ satisfying (41), $\|h_t - 1\|_{L^2(\phi)}^2 \leq \exp(-\sigma^2 \lambda ct) \|h_0 - 1\|_{L^2(\phi)}^2$. Thus, we conclude the exponential convergence of $h$ to 1. □

**Remark 8.** *The convergence of $h_t$ to 1 can be shown under more general assumption: $\beta > 0$ satisfying*

$$\lim_{T \to \infty} \int_0^T \beta(t) dt = \infty.$$

Further analysis is plausible based on the techniques in the study of partial differential equations.

**Remark 9.** *As $p_t$ is given as the convolution between $p_0$ and the Gaussian distribution, it is smooth for $t > 0$. Therefore, $h_t$ is also smooth, and the higher-order derivatives of $h_t$ are all bounded. As a consequence, the above result combined with Gagliardo–Nirenberg interpolation inequality yield that the gradient of $h$, $Dh$, and the Hessian of $h$, $D^2 h$, also converge to zero in $L^2(\phi)$.*

**Remark 10.** *Proving the uniform convergence of $h$, $Dh$, or $D^2 h$ requires an additional technical assumption that the support of $h$ is bounded. Under the assumption, another interpolation inequality, Agmon's inequality, yields the desired uniform convergence result.*

### B.2  Estimation of $L_s$

In this subsection, we investigate the estimation of $L_s$. If $J_{SM}$ is sufficiently small, then $s_\theta$ is close to $\nabla \log p_t$. We first investigate the one-sided Lipschitz constant of $\nabla \log p_t$.

**Lemma 4.** *Under the same setting as in Corollary 2, $\nabla \log p_t$ satisfies the one-sided Lipschitz condition with a constant $(-\sigma^{-2} + \|D^2(\log h)\|_\infty)$ i.e.,*

$$(\nabla \log p_t(x) - \nabla \log p_t(y)) \cdot (x - y) \leq (-\sigma^{-2} + \|D^2(\log h)\|_\infty)\|x - y\|^2. \tag{46}$$

*where $\|\cdot\|_\infty$ denotes the supremum of the matrix norm,*

$$\|D^2(\log h)\|_\infty := \sup_{x \in \mathbb{R}^d} \|D^2(\log h(x))\| = \sup_{x,y \in \mathbb{R}^d, \|y\| \leq 1} \|D^2(\log h(x))y\|. \tag{47}$$

*Proof.* From the definition of $h_t$, we have

$$\log p_t(x) = \log h_t(x) + \log \phi(x) = \log h_t(x) - \frac{x^2}{2\sigma^2} - c \tag{48}$$

for some constant $c$. As a consequence,

$$(\nabla \log p_t(x) - \nabla \log p_t(y)) \cdot (x - y) = (\nabla \log h_t(x) - \nabla \log h_t(y)) \cdot (x - y) - \sigma^{-2}\|x - y\|^2. \tag{49}$$

To prove (46), it suffices to estimate $(\nabla \log h_t(x) - \nabla \log h_t(y)) \cdot (x - y)$. From the fundamental theorem of calculus, we have

$$(\nabla \log h_t(x) - \nabla \log h_t(y)) = \int_0^1 D^2(\log h_t)(sx + (1-s)y)ds \cdot (x - y). \tag{50}$$

Using $|z^\top D^2(\log h_t(w))z| \leq \|D^2(\log h)\|_\infty \|z\|^2$ for all $w, z \in \mathbb{R}^d$, we have

$$(\nabla \log h_t(x) - \nabla \log h_t(y)) \cdot (x - y) \leq \|D^2(\log h_t)\|_\infty \|x - y\|^2 \tag{51}$$

and conclude (46). $\qquad\square$

**Remark 11.** *Based on the similar relation as in (50), the difference between $L_s$ and $-\sigma^{-2} + \|D^2(\log h)\|_\infty$ can be estimated. More precisely, we have $L_s(t) = (-\sigma^{-2} + \|D^2(\log h)\|_\infty) + \epsilon(t)$. Here, $\epsilon(t)$ depends on the difference between $s_\theta$ and $\nabla \log p_t$. Therefore, it is expected that the upper bound of $\int_0^T \epsilon(t)dt$ is given by $J_{SM}$ under suitable regularity assumptions.*

## C  Proofs of Theorem 2 and Corollary 3

For a given $t$, let

$$J_{SM}(\theta, t) := \frac{1}{2} \mathbb{E}_{p_t(x)}[\|s_\theta(x, t) - \nabla_x \log p_t(x)\|^2], \tag{52}$$

and

$$J_{DSM}(\theta, t) := \frac{1}{2} \mathbb{E}_{p_0(x(0))p_{0t}(x|x(0))}[\|s_\theta(x, t) - \nabla_x \log p_{0t}(x|x(0))\|^2]. \tag{53}$$

**Lemma 5.** *(Appendix in [36])*

$$\mathbb{E}_{p_t(x)}[\nabla_x \log p_t(x)] = \mathbb{E}_{p_0(x(0))p_{0t}(x|x(0))}[\nabla_x \log p_{0t}(x|x(0))]. \tag{54}$$

*Proof of Theorem 2.* From Lemma 5 in [36], we have

$$J_{SM}(\theta, t) = J_{DSM}(\theta, t)$$
$$+ \frac{1}{2}\left(\mathbb{E}_{p_t(x)}\left[\|\nabla_x \log p_t(x)\|^2\right] - \mathbb{E}_{p_0(x(0))p_{0t}(x|x(0))}\left[\|\nabla_x \log p_{0t}(x|x(0))\|^2\right]\right) \tag{55}$$

Note that

$$\mathbb{E}_{p_t(x)}\left[\|\nabla_x \log p_t(x)\|^2\right] - \mathbb{E}_{p_0(x(0))p_{0t}(x|x(0))}\left[\|\nabla_x \log p_{0t}(x|x(0))\|^2\right]$$
$$= \mathbb{E}_{p_t(x)}[(\nabla_x \log p_t(x))^\top (\nabla_x \log p_t(x))]$$
$$- \mathbb{E}_{p_0(x(0))}[\mathbb{E}_{p_{0t}(x|x(0))}[(\nabla_x \log p_{0t}(x|x(0)))^\top (\nabla_x \log p_{0t}(x|x(0)))|x(0)]]$$
$$= \mathrm{Var}[(\nabla_x \log p_t(x))^\top] - \mathbb{E}[\mathrm{Var}[(\nabla_x \log p_{0t}(x|x(0)))^\top]|x(0)]$$
$$+ (\mathbb{E}_{p_t(x)}[\nabla_x \log p_t(x)])^\top (\mathbb{E}_{p_t(x)}[\nabla_x \log p_t(x)])$$
$$- \mathbb{E}_{p_0(x(0))}[(\mathbb{E}_{p_{0t}(x|x(0))}[\nabla_x \log p_{0t}(x|x(0))|x(0)])^\top (\mathbb{E}_{p_{0t}(x|x(0))}[\nabla_x \log p_{0t}(x|x(0))|x(0)])]$$
$$\leq (\mathbb{E}_{p_t(x)}[\nabla_x \log p_t(x)])^\top (\mathbb{E}_{p_t(x)}[\nabla_x \log p_t(x)])$$
$$- \mathbb{E}_{p_0(x(0))}[(\mathbb{E}_{p_{0t}(x|x(0))}[\nabla_x \log p_{0t}(x|x(0))|x(0)])^\top (\mathbb{E}_{p_{0t}(x|x(0))}[\nabla_x \log p_{0t}(x|x(0))|x(0)])] \tag{56}$$

where the inequality comes from the law of total variance and our condition:

$$\mathrm{Var}[(\nabla_x \log p_t(x))^\top] - \mathbb{E}[\mathrm{Var}[(\nabla_x \log p_{0t}(x|x(0)))^\top|x(0)]]$$
$$= \mathrm{Var}[\mathbb{E}[(\nabla_x \log p_{0t}(x|x(0)))^\top|x(0)]] \tag{57}$$
$$= 0.$$

Then, we have

$$(\mathbb{E}_{p_t(x)}[\nabla_x \log p_t(x)])^\top (\mathbb{E}_{p_t(x)}[\nabla_x \log p_t(x)])$$
$$- \mathbb{E}_{p_0(x(0))}[(\mathbb{E}_{p_{0t}(x|x(0))}[\nabla_x \log p_{0t}(x|x(0))|x(0)])^\top (\mathbb{E}_{p_{0t}(x|x(0))}[\nabla_x \log p_{0t}(x|x(0))|x(0)])]$$
$$\leq (\mathbb{E}_{p_t(x)}[\nabla_x \log p_t(x)])^\top (\mathbb{E}_{p_t(x)}[\nabla_x \log p_t(x)])$$
$$- (\mathbb{E}_{p_0(x(0))}\mathbb{E}_{p_{0t}(x|x(0))}[\nabla_x \log p_{0t}(x|x(0))|x(0)])^\top (\mathbb{E}_{p_0(x(0))}\mathbb{E}_{p_{0t}(x|x(0))}[\nabla_x \log p_{0t}(x|x(0))|x(0)])$$
$$= (\mathbb{E}_{p_t(x)}[\nabla_x \log p_t(x)] - \mathbb{E}_{p_0(x(0))}\mathbb{E}_{p_{0t}(x|x(0))}[\nabla_x \log p_{0t}(x|x(0))|x(0)])^\top$$
$$(\mathbb{E}_{p_t(x)}[\nabla_x \log p_t(x)] + \mathbb{E}_{p_0(x(0))}\mathbb{E}_{p_{0t}(x|x(0))}[\nabla_x \log p_{0t}(x|x(0))|x(0)])$$
$$= 0,$$
$$\tag{58}$$

where first inequality comes from Jensen's inequality, and the last equality comes from eq. (54).

Recall that $J_{SM}(\theta, \lambda) = \int_0^T \lambda(t) J_{SM}(\theta, t) dt$ and $J_{DSM}(\theta, \lambda) = \int_0^T \lambda(t) J_{DSM}(\theta, t) dt$, $\lambda(t) > 0$, we have that $J_{DSM} \geq J_{SM}$. Plugging it in eq. (34), we can get eq. (22). $\square$

*Proof of Corollary 3.* We have $p_{0t}(x|x(0)) = \mathcal{N}(\sqrt{\bar{\alpha}_t}x(0), (1 - \bar{\alpha}_t)I)$ where $\bar{\alpha}_t = \prod_{r=1}^t (1 - \beta_t)$, which can be inferred from $p(x(t)|x(t-1)) = \mathcal{N}(\sqrt{1 - \beta_t}x(t-1), \beta_t I)$.

Thus we can show that $\mathbb{E}[\nabla_x \log p_{0t}(x|x(0))^\top|x(0)]$ is constant with respect to $x(0)$: recall $p_{0t}(x|x(0)) = \mathcal{N}(\sqrt{\bar{\alpha}_t}x(0), (1 - \bar{\alpha}_t)I)$, we have $\nabla_x \log p_{0t}(x|x(0)) = -((1 - \bar{\alpha}_t)I)^{-1}(x(t) - \sqrt{\bar{\alpha}_t}x(0))$, which is a linear function of $x(t) - \sqrt{\bar{\alpha}_t}x(0)$. Using the Gaussian density function, we have:

$$\int p_{0t}(x|x(0))\nabla_x \log p_{0t}(x|x(0))dx = 0, \tag{59}$$

As a result, $\mathrm{Var}[\mathbb{E}[(\nabla_x \log p_{0t}(x|x(0)))^\top|x(0)]] = 0$, which satisfies the condition of eq. (21) in Theorem 2.

Note that here the assumption of $f$ and $g$ is only a sufficient condition for $\mathrm{Var}[\mathbb{E}[(\nabla_x \log p_{0t}(x|x(0)))^\top|x(0)]] = 0$. In fact, any conditional distribution $p_{0t}$ that satisfies $\mathrm{Var}[\mathbb{E}[(\nabla_x \log p_{0t}(x|x(0)))^\top|x(0)]] = 0$ can lead to the same conclusion. $\square$

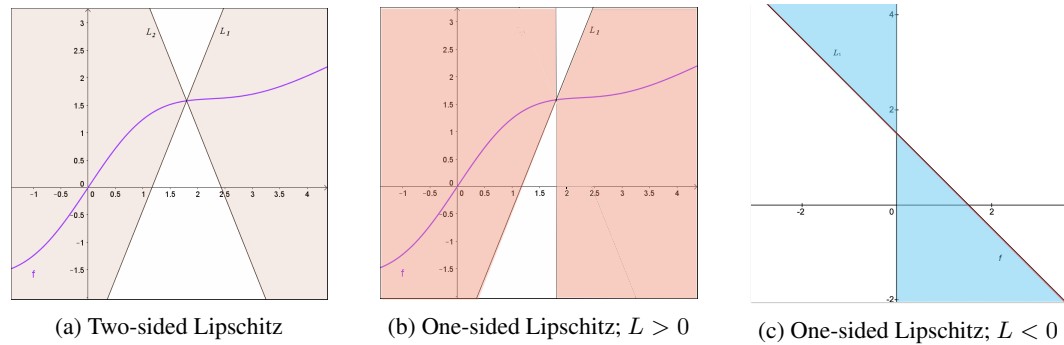

(a) Two-sided Lipschitz  (b) One-sided Lipschitz; $L > 0$  (c) One-sided Lipschitz; $L < 0$

Figure 5: Two-sided and one-sided Lipschitzness.

# D One-sided Lipschitzness

For an arbitrary Lipschitz function $F : \mathbb{R}^d \rightarrow \mathbb{R}^d$, the Cauchy-Schwarz inequality yields that

$$(F(x) - F(y)) \cdot (x - y) \leq \|F(x) - F(y)\| \|x - y\| \leq L\|x - y\|^2 \qquad (60)$$

where $L$ is the Lipschitz constant of $F$. Therefore, all Lipschitz function satisfies the one-sided Lipschitz condition:

$$(F(x) - F(y)) \cdot (x - y) \leq L\|x - y\|^2. \qquad (61)$$

As pointed out earlier in Section 3.2, the one-sided Lipschitz constant is not necessarily to be positive. For instance, if $F(x) = -ax + b$ for $a > 0$ and $b \in \mathbb{R}^d$, then $-a < 0$ can be the one-sided Lipschitz constant of $F$ while its Lipschitz constant is $a > 0$. Figure 5 visualizes this.

Note that two-sided Lipschitzness is a subset of one-sided Lipscthizness. See Figure 6 as an example.

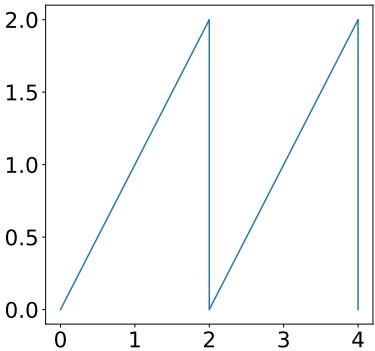

Figure 6: A function could be one-sided Lipschitz but not two-sided.

# E Full plot of $L_s(t)$ when $T = 100$

See Fig. 7b.

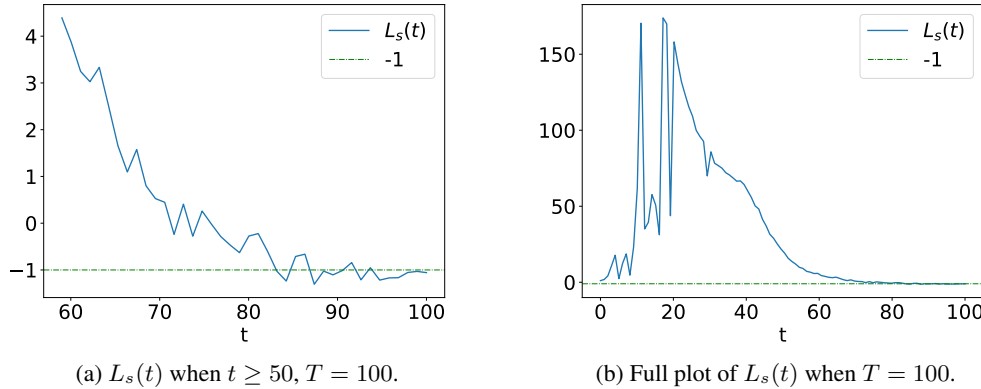

(a) $L_s(t)$ when $t \geq 50$, $T = 100$.  (b) Full plot of $L_s(t)$ when $T = 100$.

Figure 7: Plots of $L_s(t)$, $T = 100$.

## F   Numerical results on $J_{DSM}$ upper-bounding $J_{SM}$ in DDPM

To verify $J_{SM} \leq J_{DSM}$ in (22) for DDPM, we adopt the same datasets as in Fig 1, and the same training and evaluation settings in Section 4.1.

Moreover, to estimate $J_{SM}$ numerically, we estimate $p_t(x)$ by performing Gaussian kernel density estimation with bandwidth = 0.05 on sampled data. $\nabla_x p_t(x)$ is estimated by central difference approximation with interval = 0.01. The resulting plots of $J_{DSM}$ and $J_{SM}$ are shown in Fig 8, which shows that $J_{DSM}$ is an upper bound of $J_{SM}$ in DDPM during training, where $p_0$ is the dataset, and $q_0$ is the generated data distribution at the convergence of training.

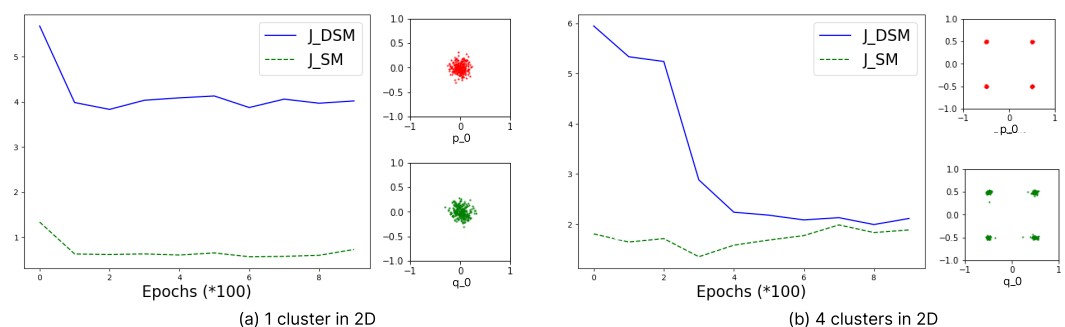

(a) 1 cluster in 2D                (b) 4 clusters in 2D

Figure 8: $J_{SM}$ and $J_{DSM}$ during training. The datasets are the same as 2D datasets in Fig 1. The training curves are obtained via training DDPM with modification of $J_{DSM}$ loss.

## G   Log-log plots with weight decay

See Fig. 9.

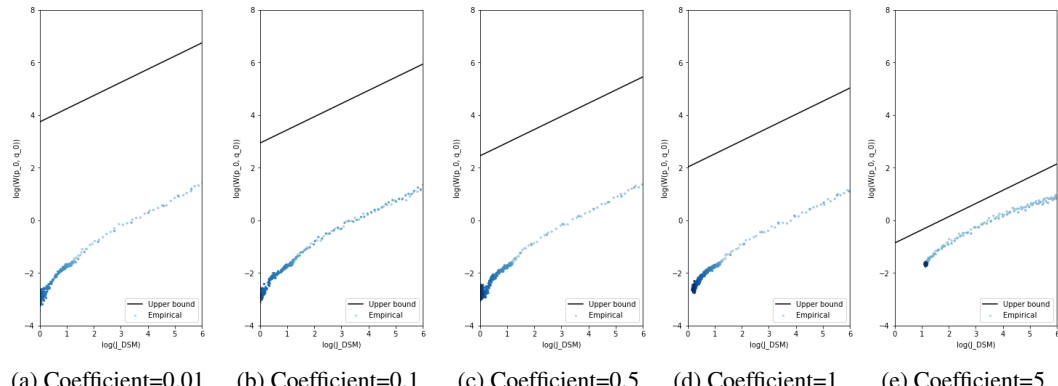

(a) Coefficient=0.01    (b) Coefficient=0.1    (c) Coefficient=0.5    (d) Coefficient=1    (e) Coefficient=5

Figure 9: Log-log plots for different weight decay coefficients. As the weight decay coefficient increases, the theoretical upper bound is approaching the empirical one.