# OpenReview forum: "Score-based Generative Modeling Secretly Minimizes the Wasserstein Distance"
_NeurIPS.cc/2022/Conference — NeurIPS 2022 Accept_

### Official Review · Reviewer_EPR5 · 2022-07-05

**Rating:** 5
**Confidence:** 3
**Soundness:** 2 fair
**Presentation:** 2 fair
**Contribution:** 2 fair

**Summary:**

This paper shows that score-based models minimize the Wasserstein distance between them and that the Wasserstein distance is upper bounded by the square root of the objective function up to multiplicative constants and a fixed constant offset.

**Questions:**

1. The main question is whether or not the theorems in this paper lead to the finding that score-based models minimize the Wasserstein distance between them, i.e., $W_2(p_0,q_0) = \min W(p,q) = 0$. The theorems in this paper are based on the upper bounds of $W_2$. Remark 1 indicates that Corollary 1 implies that $W_2$ converges to $0$ under some assumptions. Hence, it is not guaranteed that score-based models minimize the Wasserstein distance between them.
2. The assumptions in the main theorems seem to be strong. For example,
- Remark 1: the conditions $J_{SM} \to 0$, $p_T = q_T$, and $\lambda(t) = g(t)^2$ seem to be strong. Provide some practical examples satisfying all the conditions.
- Theorem 2: Provide some practical examples of $p_{0t}$ satisfying (15).
3. As a result, I cannot understand what the contribution of this paper is and why the results in this paper are significant. In particular, I cannot understand why the upper bounds of $W_2$ are needed for deep machine learning and its related areas.

**Limitations:**

1. There is a gap between the claim in the abstract and the main theorems in this paper.
2. The assumptions in this paper seem to be strong.
3. I do not know that numerical results in section 4 are appropriate since the condition $W_2(p_T, q_T) = 0$ is assumed. Also, I do not know why the setting in Page 6 (e.g., a 3-layer neural network, AdamW with lr = 0.001) are appropriate. The numerical results are based on factitious setting, and hence, limited results.

**Strengths And Weaknesses:**

Strengths: This paper shows that score-based models minimize the Wasserstein distance between them and that the Wasserstein distance is upper bounded by the square root of the objective function up to multiplicative constants and a fixed constant offset.

Weaknesses:
1. The assumptions in the main theorems seem to be strong.
2. The abstract indicates the claim such that score-based models minimize the Wasserstein distance between them. However, the main theorems cannot follow the claim.

---

> ### Author Response · Authors · 2022-08-02
> **Response**
>
> We appreciate your thoughtful comments, which helped us to improve the manuscript. While a general convergence result remains still open, as discussed in Section 6, **we added a new Corollary 2 in the case of DDPM.** This was briefly illustrated in the appendix of the original manuscript. We further improved the statement based on the study of the upper bound.
>
> > >**Corollary 2**. Under the same setting as in Corollary 1, let $f(t) = -\frac{\beta(t)}{2} x$, $g(t) = \sigma \sqrt{\beta(t)}$, and $q_T = \phi \sim \mathcal{N}(0, \sigma^2 I)$. Then, we have
>
> \begin{equation}
> W_2(p_0, q_0) \leq \sqrt{2 \left(\int_0^T g^2(t) I(t)^2 dt \right) J_{SM}} + \exp\left(\int_0^T \beta(r) L_s(r) dr\right) W_2(p_0, \phi).
> \end{equation}
>
> Here, $\sigma>0$, and $\beta : [0, T] \rightarrow (0, +\infty)$ is a positive function.
>
> In addition to Corollary 2, **further analysis on the one-sided Lipschitz constant $L_s$ has been done in Appendix H** (Lemmas 4 and 5 and Remarks 8-11). As the arguments are technical, we provide a formal analysis in Remark 4 immediately after Corollary 2 at the end of Section 3.1.
>
> > >**Remark 4.** If $s_\theta$ is close enough to $\nabla \log p_t(x)$, then $L_s(t) \sim (-\sigma^{-2} + \| D^2 (\log h_t) \|_\infty)$
> Here, $h_t:= p_t/\phi$ exponentially converging to 1 as $t \rightarrow \infty$ (see Lemma 4) and $D^2 (\log h_t)$ is the Hessian of $\log h_t$. Thus, $L_s(t)$ converges to $-\sigma^{-2}$, as observed in our experiments in Fig. 4a when $\sigma = 1$. Consequently, the second term in the upper bound in Corollary 2 decays as $T$ increases (see Fig. 4b).
>
> **These imply that the offset can be ignored under certain conditions, which are achievable (e.g., DDPM with large T).** Below, we provide the responses corresponding to each of the reviewer's comments.
>
> **`Q. Minimization of the Wasserstein distance`**
>
> >The main question is whether or not the theorems in this paper lead to the finding that score-based models minimize the Wasserstein distance between them, i.e., $W_2(p_0, q_0) = \min W_2(p, q) = 0$.
>
> The presence of the second term $I(T)W_2(p_T, q_T)$ is inevitable in the structure of the score-based model. In other words, if the final distribution $p_T$ of the forward process is too different from a known prior distribution $q_T$ in the reverse process, it is unlikely that the data distribution $p_0$ is close to the model distribution $q_0$. Similarly, in [32], the upper bound of the KL divergence from $p_0$ to $q_0$ is given as the addition of $J_{SM}$ and the KL divergence from $p_T$ to $q_T$.
>
> [32] Y. Song, C. Durkan, I. Murray, and S. Ermon. Maximum likelihood training of score-based diffusion models. Advances in Neural Information Processing Systems
>
> >The theorems in this paper are based on the upper bounds of $W_2$. Remark 1 indicates that Corollary 1 implies that $W_2$ converges to 0 under some assumptions. Hence, it is not guaranteed that score-based models minimize the Wasserstein distance between them.
>
> However, we agree with the reviewer's point that the assumption $p_T = q_T$ can be too strong. **Without assuming $p_T=q_T$, we added more analysis about the offset $I(T)W_2(p_T,q_T)$ for DDPM in the updated Remark 4, which also justified the choice of SDE in DDPM using our theory**.
>
> The main idea is that in DDPM, $p_T|p_0 \sim \mathcal{N}(\sqrt{\prod_{t=1}^T(1-\beta_t)},(1-(\prod_{t=1}^T(1-\beta_t)))I)$ and when $T$ is large $T$, $p_T$ would approach $\mathcal{N}(0,I)$ as long as $\beta_t<1-\epsilon, \forall t$, which is satisfied generally since $\beta_t$ is chosen to be small in practice. We can utilize the log concavity of this Gaussian distribution to bound our $L_s$ when minimizing $J_{SM}$: since $s_\theta$ would approach $\nabla \log p_t$ during training, we can expect that $L_s(t)$ would be close to $-1$ when $t$ is large, which would avoid an explosion in $I(T)$ when $T$ is sufficiently large. **As a result, the offset would decay almost exponentially when $T$ is large in DDPM, which is verified in our updated experiments (Section 4.3.2 and Fig.4 in blue)**.
>
> > There is a gap between the claim in the abstract and the main theorems in this paper.
>
> We agree with the reviewer that the gap exists in the original manuscript. Our further analysis and experiments above show that the Wasserstein distance is minimized under suitable conditions that are achievable (e.g. DDPM with large T), which supports our claim in the abstract.

---

> > ### Author Response · Authors · 2022-08-02
> > **Response (cont.)**
> >
> > **`Q. Other assumptions`**
> > > The assumptions in the main theorems seem to be strong. For example,
> > Remark 1: the conditions $J_{SM} \rightarrow 0$, $p_T = q_T$, and $\lambda(t) = g(t)^2$ seem to be strong. Provide some practical examples satisfying all the conditions.
> >
> > * $\lambda(t) = g(t)^2$: Remark 2 addresses this assumption. For clarity, we added Corollary 4 in Appendix A in the updated manuscript.
> >
> > > > **Remark 2.** If $g^4 I^2 \lambda^{-1}$ is integrable in $[0,T]$, the similar inequality as in the inequality in Corollary 1 holds for other choices of $\lambda$. See Corollary 4 in Appendix A.
> >
> > * $J_{SM} \rightarrow 0$: The main idea of score-based models is to find $s_\theta$ close enough to $\nabla \log p_t$. Furthermore, Theorem 2 yields $J_{SM} \leq J_{SM}$ where $J_{DSM}$ is a loss function of the model. Therefore, as we train the model, it is expected that $J_{SM}$ converges to zero. In this sense, we do not agree with the reviewer's claim that the assumption $J_{SM} \rightarrow 0$ is too strong.
> >
> >
> >
> > >Theorem 2: Provide some practical examples of $p_{0t}$ satisfying (15).
> >
> > It was already presented in Corollary 3. Diffusion Probabilistic Models (DDPM) satisfy the condition (15). It is stated in the proof of Corollary 3, but we also added the statement before Corollary 3 for clarification.

---

> > ### Comment · Reviewer_EPR5 · 2022-08-05
> > **Your replies on my comments**
> >
> > Thank you for your valuable comments and the revised manuscript. I would like to check the revised manuscript to understand your claims.  Hence, I require sufficient time for checking the revised manuscript.

---

### Official Review · Reviewer_v4Sv · 2022-07-05

**Rating:** 6
**Confidence:** 4
**Soundness:** 3 good
**Presentation:** 3 good
**Contribution:** 3 good

**Summary:**

This article uses optimal transport theory to prove that score-based generative models not only minimize the Kullback-Leibler divergence but also the Wasserstein distance between the generated samples distribution and the data distribution. In addition, it establishes that the Wasserstein distance can be upper-bounded by the square root of the objective function (up to multiplicative and additive constants) for both score-based models and denoising diffusion models.
This paper then discusses the impact of weight decay and the number of timesteps $T$ on the upper bounds. Theoretical results are experimentally checked and illustrated on simple synthetic datasets. Beyond that, the article also discusses the robustness of score-based models to noise perturbation on the datasets using the Wasserstein distance.

**Questions:**

- The article discusses casts light on reducing the intercept $I$ by applying weight decay (since it allows to reduce the Lipschitz constant $L_s$). Considering that $I$ also depends on the SDE used through $L_f$ and $g^2$, did the authors study the effect of the choice of the SDE on the intercept value given a fixed dataset?
- On figure 6 (Appendix E), we can notice that the denoising score matching loss does not decrease much during training. Can the authors explain why the observed gap is equal to the theoretical one and not the consequence of the neural network of the denoising score matching model underfitting the distribution? (since it is a very small network).


**Limitations:**

There is no direct potential negative societal impact of the article proposed in this article.

[Edit] - I raised my score

**Strengths And Weaknesses:**

- The results proved in this article bring new interpretations on the score-based models learning properties, which were, to the best of our knowledge, not known before. The results are clearly stated and discussed thanks to numerous remarks sections.
- The article is well written and easy to follow, despite some paragraphs that could be made clearer. Previous works, both in the field of generative models and optimal transport, are referenced and well-integrated in the narration.
Notations are clearly defined and respected, even if prior knowledge about the related works is sometimes needed. Also, note that objective functions $J_{SM}$ $J_{DSM}$ are referenced (line 59) before definitions, or that the variables with respect to which variance and expectation are taken in Eq. 15 should be precised.
- The experiments and figures help with the comprehension of the article and provide illustrations of the theoretical statements.

- The motivation of the article could be stated more clearly. Besides the theoretical result, the article could mention the benefits of minimizing the Wasserstein distance between the model and the data distributions.
- My biggest concern is about the Experiments part: This is mostly a theoretical paper and the presented experiments look somewhat superficial or confusing. The fact that the upper bound of Eq. 19 depends on the training procedure is rather unusual. This could have been controlled before training by devising a dedicated architecture. There are numerous resources on this subject, so that controlling the network's Lipschitz constant only via weight regularization does not seem convincing in this case. Also, it would have been interesting to discuss if the upper bound of Eq. 19 could suggest new ways to select the forward SDE. In the paper, only the dependence on the network's Lipschitz constant $L_s$ is discussed.
The assumption $W_2(p_T, q_T)=0$ depends on the choice of the forward SDE and is only approximately true. Completely removing it from Eq. 19 does not show the balance between low Wasserstein distance at $T$ and small $I(t)$.
- It is not very clear how the theoretical results of this article can motivate improvement in the case of score-based model training on more complex datasets in practice.
- Some details could be added regarding the training and evaluation of the network: T=10 timesteps seems like a too small value compared to what is regularly used in the literature. The number of samples used to compute $W_2$ is also missing.
- Some adjustments regarding the overall organisation of the paper could be made: "The score matching and diffusion probabilistic models" subsection in §2.4 could be included in the Introduction. This could help understanding that there is not much distinction between score-based generative models and DDPMs, as suggested in the Introduction, and would help understanding Sect. 2. The sketch of the proof could follow the theorem.

- l. 219 $L_s(t)$ instead of $L_s(r)$

---

> ### Author Response · Authors · 2022-08-02
> **Response**
>
> **`Q. Controlling the network's Lipschitz constant only via weight regularization does not seem convincing...`**
>
> We sincerely thank your suggestion here! As per your suggestion, we tried different ways to control the Lipschitz constant. Specifically, we tried spectral normalization (which is used in iResNet[1] and Spectral normalized GAN[2]), and weight clipping (which is used in WGAN[3]). Our current preliminary results are added to Section 4.3.1 in blue. We observe that both techniques can effectively control the Lipschitz constant hence tighten our upper bound. However, we have not observed any sigfinicant difference between weight decay and these new approaches. We will continue exploring various approaches to control the Lipschitz constant and include a more extensive study in the final version.
>
>
> *References.*
> * [1] J. Behrmann, W. Grathwohl, R. T. Chen, D. Duvenaud, and J.H. Jacobsen. Invertible residual networks. In International Conference on Machine Learning, pages 573–582. PMLR, 2019.
> * [2] Takeru Miyato, Toshiki Kataoka, Masanori Koyama, Yuichi Yoshida. Spectral Normalization for Generative Adversarial Networks, ICLR 2018
> * [3] M. Arjovsky, S. Chintala, and L. Bottou. Wasserstein generative adversarial networks. In International conference on machine learning, pages 214–223. PMLR, 2017.
>
>
> **`Q. If we remove the second term in the upper bound (I*W_2), one cannot capture the balance between the two terms. How can we justify this?`**
>
> In Appendix G of the original submission (now Appendix H in the revision), we briefly mentioned that $W_2(p_T,q_T)$ would exponentially decay, but did not fully justify whether the product of $I$ and $W_2$ also vanishes or not.
>
> To better justify our omission of the $I \times W_2$ term, we added an in-depth analysis of $I\times W_2$ in our revision. In Remark 4 of the revision, we provide a new upper bound on $I(T)*W_2(p_T,q_T)$ for DDPM models.
>
> > [Upper bound on $I(T)*W_2(p_T,q_T)$ for DDPM models]
> > \begin{equation}
> I(T)*W_2(p_T,q_T) \leq C \exp\left(\int_0^T \beta(r) L_s(r) dr\right)
> \end{equation}
> where $\beta : [0, T] \rightarrow (0, +\infty)$ is a positive function
>
> Although the exact analysis of $L_s$ is still open, one can still check the upper bound numerically. **We conjecture that the above upper bound exponentially decays at least in DDPM for the following reason**:
>
> For DDPM, $p_T|p_0 \sim \mathcal{N}(\sqrt{\prod_{t=1}^T(1-\beta_t)},(1-(\prod_{t=1}^T(1-\beta_t)))I)$ and when $T$ is large $T$, $p_T$ would approach $\mathcal{N}(0,I)$ as long as $\beta_t<1-\epsilon, \forall t$. Due to the log concavity of the Gaussian distribution, we can bound $L_s$: since $s_\theta$ would approach $\nabla \log p_t$ when $J_{SM}$ is minimized, we can expect that $L_s(t)$ would be close to $-1$ when $t$ is large. **We numerically computed $L_s(t)$ for larger T in Section 4.3.2 and Fig.4 in our revision, supporting our conjecture**. Rigorously proving this conjecture is an open problem.
>
>
> **`Q. How does the choice of the SDE affect I(t)?`**
>
> See the discussion above.
>
>
> **`Q. The denoising score matching loss does not seem to decrease much. Is it underfitting?`**
>
> We confirm that it is not underfitting. We updated the plots with generated data at convergence. **The updated plots are marked as blue in Appendix E**.
>
> **`Q. T=10 seems too small. Also, number of samples to compute W is not specified.`**
>
> We added more experiments with larger T in Section 4.3.2. The number of samples used to compute W is 512 samples from respective distribution.
>
> **`Q. Some adjustments regarding the overall organisation...`**
>
> Thanks for your suggestions! We added our organization of the proof and main ideas at the beginning of Section 3.

---

> > ### Author Response · Authors · 2022-08-08
> > **Reply to Reviewer v4Sv**
> >
> > Thank you for your constructive feedback. Since the discussion period is ending soon, we wanted to check in and ask if the rebuttal sufficiently answered your queries. We are happy to address if there are any further questions.

---

### Official Review · Reviewer_QQUJ · 2022-07-12

**Rating:** 7
**Confidence:** 1
**Soundness:** 3 good
**Presentation:** 3 good
**Contribution:** 3 good

**Summary:**

- This paper discusses properties of the PDF realized by the data generating process of score-based models (denoising diffusion models), which have received a great deal of attention recently.
The three main theorems are as follows
  1. If the true density is $p_0$ and the density of generated data is $q_0$, then its Wasserstein2 distance $W_2(p_0, q_0)$ is bounded above by
$$W_2(p_T, q_T) \times \text{(a constant dependent on Lipschitz constants)} + \sqrt{\text{(training loss)}} \times \text{const}.$$
Therefore, under ideal conditions such that $p_T=q_T$, the loss function is an upper bound of the Wasserstein2 distance between the true and generated densities.
  2. The loss function commonly used in the training of score-based model is just an approximation of the true score-matching loss. In this paper, a theorem that gives the relation betweem them is provided. This gives a theoretical backborn  that the approximate loss function used in practice can be used as a surrogate function for the true objective function which is difficult to evaluate exactly.
  3. Perturbing around the initial value (eliminating singularities at the time origin) is a fundamental technique in practical situations. A theorem is provided to evaluate the upper bound of the Wasserstein distance between the true density and the perturbed density. This is a direct consequence of the triangular inequality of the Wasserstein distance and the above theorem.

- It is then demonstrated numerically using a toy model and toy data that the upper bound given by these theorems actually valid. It is also argued that the main factors of the gap between the upper bound and the loss are related to the the Lipschitz constant of the neural network.

- The proof begins with a review of the fundamentals that the Fokker-Planck equation is analogous to the equation of continuity (or the law of mass conservation) which is the fundamental relationship between density and velocity field. Based on that argument, a equation is presented that states that the time derivative of the Wasserstein2 distance is expressed by the velocity field, the proof of which is based on outside literature.


**Questions:**

- The discussion in this paper is based on a continuous stochastic process, but the numerical simulations appear to be dealing with discretized processes with the step size of $t = 1$. Is my understanding correct that that this is justified by Cor 2?

- (The issue that interested me most while reading the text was the behavior in higher dimensions rather than the 2-dim toy data. However, this was discussed in the appendix, albeit to a limited extent.)


**Limitations:**

- It is argued in section 6 that more general conditions make theoretical evaluation more difficult.


**Strengths And Weaknesses:**

- These theorems are interesting because they guarantee fundamental properties of the score-based generative models. (However, I did not follow all the technical details of the proofs, and am not still confident in my understanding.)

- It is an interesting claim that there is a theoretical guarantee that the same algorithm minimizes KL and Wasserstein distances simultaneously, as opposed to GAN, which leads to different algorithms depending on its distance measure.

- These evaluations are strictly valid only for special cases, and for more general cases, theoretical evaluation is an open question. This is discussed in Section 6.

---

> ### Author Response · Authors · 2022-08-02
> **Response**
>
> We sincerely appreciate your interest and the detailed summary of our manuscript. In particular, we are glad to hear the following two points.
>
> >These theorems are interesting because they guarantee fundamental properties of the score-based generative models. (However, I did not follow all the technical details of the proofs, and am not still confident in my understanding.)
>
> >It is an interesting claim that there is a theoretical guarantee that the same algorithm minimizes KL and Wasserstein distances simultaneously, as opposed to GAN, which leads to different algorithms depending on its distance measure.
>
> All the raised comments and questions have been considered below.
>
>
> **`Q. Analysis assumes continuous stochastic process, but simulations deal with discretized process. Is it justified in the current paper?`**
>
> Thanks for a great question. The reviewer is correct that our current analysis is based on continuous-time approximation, which can be viewed as time-scale asymptotic analysis. We have not justified this approximation. However, the same approximation is farily standard and has been used in the theoretical studies in the field, e.g., (Song et al., 2021).
>
> That being said, we agree that a rigorous "translation" of theoretical results is an important research direction, as studied for Stochastic Normalizing Flows (Hodgkinson et al., 2020). It is one of our ongoing research directions. In particular, we plan to use the numerical techniques that were developed to bridge the gap between discrete-time dynamic systems and their continuous-time analogous.
>
> *References.*
> * Song, Y., Durkan, C., Murray, I., & Ermon, S. (2021). Maximum likelihood training of score-based diffusion models. Advances in Neural Information Processing Systems, 34, 1415-1428.
> * Hodgkinson, L., van der Heide, C., Roosta, F., & Mahoney, M. W. (2020). Stochastic normalizing flows. arXiv preprint arXiv:2002.09547.
>
> **`Q. Limited experimental results for high dimension cases`**
>
> Evaluating our upper bounds for high dimensional cases requires an efficient algorithm to compute (or upper bound) one-sided Lipschitz constant, but there is no known polynomial algorithm that can do so to the best of our knowledge. We believe that this is a very interesting algorithmic open question.

---

> > ### Author Response · Authors · 2022-08-08
> > **Reply to Reviewer QQUJ**
> >
> > Thank you for your constructive feedback. Since the discussion period is ending soon, we wanted to check in and ask if the rebuttal sufficiently answered your queries. We are happy to address if there are any further questions.

---

> > ### Comment · Reviewer_QQUJ · 2022-08-08
> > **Thank you for your reply**
> >
> > Thank you for answering my questions.
> >
> > I understood that the present study deals with an ideal model of the continuous limit, and how to approximate this limit in a discrete manner can be another issue, which is an open question to be addressed in the next step using techniques of numerical analysis, dynamical systems, or others. Personally I feel more comfortable with the scenario "approximating a continuous physical system with a discrete model" than the opposite one, so this scenario makes sense to me.
> >
> > In any case, understanding the behavior in some limit cases (continuous limit in this paper) will provide some useful insights and information for developing practical methods. So the present results will be interesting as a guideline for future studies in this area. I have not yet fully followed the details of the proofs, but if these are correct, they would be of sufficient interest.

---

### Official Review · Reviewer_Y7JZ · 2022-07-13

**Rating:** 8
**Confidence:** 2
**Soundness:** 3 good
**Presentation:** 2 fair
**Contribution:** 4 excellent

**Summary:**

This is a theoretical paper on score-based generative models. The main result states that the Wasserstein distance between the data distribution and the generated distribution of such models is upper bounded by the square root of their objective function up to multiplicative constants and a fixed constant. This suggests that such models tend to minimize the Wasserstein distance between the target distribution and the estimated one, and complements another recent similar result on the Kullback-Leibler divergence [31]. Another theorem connecting this result to diffusion models is also provided. The findings are supported by numerical experiments on toy data (Gaussian mixtures).

**Questions:**

- The analysis is conducted in the continuous time domain, whereas in practice, diffusion model are always trained using a discrete, finite number of time step. A comment on how this theoretical analysis translates to the discrete setting would be welcome.
- In a similar vein, it is not clear how the various continuous-time integrals were numerically calculated in practice by the authors in the various experiments.
- The loss function defined in eq. (10) is not very well explained or motivated.
- Please check the bibliography, it contains many mistakes. For instance, bibliographical reference [31] appears to be incorrect (it seems to include a long list of "editors"). Moreover, many bibliographical references are duplicate, eg., [31] and [32],  [33] and [34], [36] and [37].

Typos:
- L29: the upper bound => an upper bound
- L59: neither variables J_SM or J_DSM have been properly defined at this point of the paper
- L94: from in the above => from the above
- L269: makes us hard => makes it hard


**Limitations:**

The limitations of the proposed theoretical analysis are adequately addressed in Section 6.

**Strengths And Weaknesses:**

## Strength
- The paper provides a strong contribution towards a better theoretical understanding of the dramatic empirical success demonstrated by score-based and diffusion-based generative models in recent years.

## Weakness
- The paper is hard to follow for a reader who is not deeply familiar with PDE formulations of scored-based models, diffusion models, and Wasserstein metrics.

Unfortunately I do not have this background nor the time to acquire it, and hence was not able to check the details of the proofs in the paper. My assessment of the paper is based on the assumption that the proofs are all correct.

---

> ### Author Response · Authors · 2022-08-02
> **Response**
>
> We greatly appreciate the reviewer's acknowledgment that our work provides a strong contribution to a better theoretical understanding of score-based and diffusion-based generative models. All the raised comments and questions have been considered below.
>
> **`Q. How does "continuous-time" analysis translate to practical settings where discrete time steps are used?`**
>
> Thanks for a great question. The reviewer is correct that our current analysis is based on continuous-time approximation, which can be viewed as time-scale asymptotic analysis. The same approximation is farily standard and has been used in the theoretical studies in the field, e.g., (Song et al., 2021).
>
> That being said, we agree that a rigorous "translation" of theoretical results is an important research direction, as studied for stochastic normalizing flows (Hodgkinson et al., 2020). It is one of our ongoing research directions. In particular, we plan to use the numerical techniques that were developed to bridge the gap between discrete-time dynamic systems and their continuous-time analogous.
>
> *References.*
> * Song, Y., Durkan, C., Murray, I., & Ermon, S. (2021). Maximum likelihood training of score-based diffusion models. Advances in Neural Information Processing Systems, 34, 1415-1428.
> * Hodgkinson, L., van der Heide, C., Roosta, F., & Mahoney, M. W. (2020). Stochastic normalizing flows. arXiv preprint arXiv:2002.09547.
>
> **`Q. How were the various continuous-time integrals numerically calculated in the experiments?`**
>
> We approximated each integral as a discrete sum with small time steps. We added these implementation details to the revision.
>
> **`Q. Explanation of eq. (10)`**
> > The loss function defined in eq. (10) is not very well explained or motivated.
>
> Thank you for your great comment. The definition in eq. (10) requires a bit of technical details -- simply speaking, it naturally arose from the estimation of the Wasserstein distance. Based on your feedback, we simplified the flow and rewrote our main theorem (Theorem 1) such that we do not have to introduce this technical definition anymore.
>
> **`Q. Bibliography and typos`**
>
> All raised comments on the bibliography and typos have been addressed as suggested.

---

> > ### Comment · Reviewer_Y7JZ · 2022-08-03
> > **Thanks**
> >
> > I thank the authors for replying to my questions and updating the paper accordingly. I leave my overall score of 8 unchanged.

---

### Author Response · Authors · 2022-08-02
**General Comments to AC and All Reviewers**

(R1 - Y7JZ, R2 - QQUJ, R3 - v4Sv, R4 - EPR5)

We would like to thank all reviewers for their thoughtful comments and remarks. We greatly appreciate the reviewers' acknowledgment that our work is **novel with a soild contribution (R1, R2, R3)**. In our work,
1. We provide a new theoretical analysis of the score-based generative models. Our work guarantees that the same algorithm minimizes the KL divergence and Wasserstein distances simultaneously, as opposed to other generative models such as GANs;
2. We prove that the approximated loss function used in practice can be a valid upper bound of the theoretical objective function which is generally intractble;
3. We verify our theoretical upper bound via numerical experiments.


We are glad that reviewers think our paper **well-written (R2, R3) and the experiments support our theoretical findings (R1, R2, R3)**, and we appreciate the time and effort that they had put into their reports.

We also sincerely thank reviewers who raise extra questions about **the offset ($I(T)W_2(p_T,q_T)$) analysis in our main theorem (R3, R4)**, which encourage us to work on deeper understanding about this offset **both theoretically and empirically**. **The major updates are included as follows**:

1. **We claim that this offset can be ignored under certain conditions (e.g. DDPM with large T), and these conditions are generally satisfied in practice and also support the design choice of DDPM** via our updated Corollary 2 and Remark 4, with more theoretical analysis in Appendix H.
2. **We verify this claim via experiments in Section 4.3.2**: when T is large (e.g., T=100), we plotted $L_s$ at the convergence of training diffusion models, which converges to $-1$ as $t \rightarrow T$ as shown in Fig. 4(a), verifying our Remark 4. Moreover, we plotted $I(T)W_2(p_T,q_T)$ for different choices of $T$ at the convergence of training, showing that it decays at a exponential rate approximately in practice.
3. **We add extra experiments on different ways of controlling the Lipschitz constant in Section 4.3.1**: we adopt spectral normalization (used in iResNet and spectral normalized GAN) and weight clipping (used in WGAN) except for our original weight decay reguarlization, and our conclusion about the relation between controlling $L_s$, the intercept and the minimal loss remains the same.

We hope above changes can answer some common questions from the reviewers. In order to ease the revision, in the updated manuscript **the most important changes are marked in blue**. We also respond to indivial comments from each reviewers below.

---

### Meta-Review · Area_Chair_mJGz · 2022-08-29

**Recommendation:** Accept
**Confidence:** Certain

**Metareview:**

This paper considers a reverse process (4) of the diffusion process (1), and provides an upper bound (Theorem 1) of the Wasserstein distance $W_2(p_0,q_0)$ of a data distribution $p_0$ and the distribution $q_0$ defined by the reverse process at time $t=0$, in terms of the approximation error $J_I$ of the score function $\nabla\log p_t$ by $s_\theta(t)$ and the Wasserstein distance $W_2(p_T,q_T)$. It implies that, provided that $W_2(p_T,q_T)$ is zero, approximating the score function by $s_\theta$ via training will make $W_2(p_0,q_0)$ small, allowing generative modeling of $p_0$ by $q_0$. A more practical upper bound (Theorem 2), as well as an analysis on effects of perturbations of the data distribution $p_0$ (Theorem 3), is also provided.

The review ratings/confidences were 8/2, 7/1, 6/4, and 5/3. The average is above the acceptance threshold. Upon reading the reviews, the author responses, as well as the paper itself, I found that the comments by the reviewers with lower ratings were mostly reasonable given the initial version of this paper which was difficult to follow in some places, as well as the fact that one would need simplifications like the continuous-time formulation and assumptions to guarantee integrability of relevant quantities, for theoretical development like the one conducted in this paper, but I found that most of them, especially the issue of the competition between $I(T)$ and $W_2(p_T,q_T)$ in the case of the Denoising Diffusion Probabilistic Models as summarized in Remark 4 in the revised version, were properly addressed in the revised version. I would therefore like to recommend acceptance, which would encourage further discussion among the attendees of the conference on more technical details.

A few minor points:
- Notational inconsistency: The authors used the notation $g^2(.)$ to mean $(g(.))^2$ and $I(.)^2$ to mean $(I(.))^2$, which seem inconsistent to me.
- Equation (11): The integrand $L_f(r)+L_s(r)g^2(r)$ should be put in parentheses.
- The term $\frac{1}{2}\log2$, which comes from the prefactor 2 in the square root in Equation (16), should be added to the right-hand side of Equation (19).

**Award:**

No

---

### Decision · Program_Chairs · 2022-09-14

Accept